# Mammalian genes induce partially reprogrammed pluripotent stem cells in non-mammalian vertebrate and invertebrate species

**Ricardo Antonio Rosselló[1,2,3]\*, Chun-Chun Chen[2,3], Rui Dai[2], Jason T Howard[2,3], Ute Hochgeschwender[2,4]\*, Erich D Jarvis[2,3]\***

[1]Department of Biochemistry, University of Puerto Rico Medical Sciences Campus, San Juan, Puerto Rico; [2]Department of Neurobiology, Duke University Medical Center, Durham, United States; [3]Howard Hughes Medical Institute, Duke University Medical Center, Durham, United States; [4]Duke Neurotransgenic Laboratory, Duke University Medical Center, Durham, United States

**Abstract** Cells are fundamental units of life, but little is known about evolution of cell states. Induced pluripotent stem cells (iPSCs) are once differentiated cells that have been re-programmed to an embryonic stem cell-like state, providing a powerful platform for biology and medicine. However, they have been limited to a few mammalian species. Here we found that a set of four mammalian transcription factor genes used to generate iPSCs in mouse and humans can induce a partially reprogrammed pluripotent stem cell (PRPSCs) state in vertebrate and invertebrate model organisms, in mammals, birds, fish, and fly, which span 550 million years from a common ancestor. These findings are one of the first to show cross-lineage stem cell-like induction, and to generate pluripotent-like cells for several of these species with in vivo chimeras. We suggest that the stem-cell state may be highly conserved across a wide phylogenetic range.

**\*For correspondence:** ricardo.rossello@upr.edu (RAR); ute@neuro.duke.edu (UH); jarvis@neuro.duke.edu (EDJ)

**Competing interests:** The authors declare that no competing interests exist.

## Introduction

Stem cells are in an early undifferentiated state and have the potential to differentiate into a variety of cell types and tissues, both in vitro and in vivo, including in developing embryos and grafted adult tissues (*Badylak et al., 2012*). Accordingly, stem cells provide a powerful platform to study development (*Arendt, 2008*), tissue regeneration (*Langer and Vacanti, 1999*; *Rosselló et al., 2009*), disease mechanisms (*Colman and Dreesen, 2009*), and gene therapeutic approaches to the brain and other organs (*Hwang et al., 2011*). Embryonic stem cells (ESCs) have the potential to be differentiated to most if not all cell types (pluripotent), whereas more differentiated stem cells, such as those in the skin, have a more restricted differentiation potential (multipotent or unipotent) (*Collas et al., 2007*). Induced pluripotent stem cells (iPSCs) are once mature cells that have been de-differentiated to become like the embryonic state (*Thomson et al., 1998*; *Takahashi and Yamanaka, 2006*; *Takahashi et al., 2007*; *Yu et al., 2007*; *Maherali and Hochedlinger, 2008*; *Stadtfeld et al., 2008*). One major advantage of iPSCs is that they can be made from differentiated cells, such as skin or fibroblasts, of an individual and do not require isolating cells from 2–6 day old embryos, which is controversial for human studies (*Lo and Parham, 2009*). The finding that simple over-expression of four genes (*Oct4*, *Sox2*, *Klf4* and *c-myc*) was sufficient to generate iPSCs from adult cells of mice (*Takahashi and Yamanaka, 2006*) and humans (*Takahashi et al., 2007*; *Yu et al., 2007*; *Sommer et al., 2009*) made the process of generating and studying stem cells much more tractable in certain other mammalian species,

**eLife digest** Stem cells are 'pluripotent'—in other words, they have the potential to become many other cell types. This ability makes them extremely valuable for research. They also hold substantial promise for medical applications, since they can be used to replace cells lost or damaged by disease or injury.

Embryos represent a rich source of stem cells; however, obtaining these cells from human embryos raises obvious ethical and practical concerns, and they have also been difficult to isolate from many species. A recent discovery circumvented these issues for humans and several mammalian species commonly studied in the laboratory. This technique can turn cells from adult mammals into 'induced pluripotent stem cells', or iPSCs, by switching on four genes. Nevertheless, no analogous method has yet been established to create similar cell populations in non-mammalian organisms, which are also important models for human development and disease.

Now, Rosselló et al. have shown that cells from both invertebrate and non-mammalian vertebrate species—including birds, fish and insects—can be reprogrammed into cells that closely resemble iPSCs. Intriguingly, these cells were created by switching on the same four genes that generate iPSCs in mammals, even though vertebrates and invertebrates are separated by around 550 million years of evolution.

Rosselló et al. used a viral vector that carries the four stem-cell genes (from the mouse) into target cells from the different species. The genetically altered cells developed into iPSC-like cells with many of the characteristics of natural mammalian and bird stem cells. To confirm that the cells were pluripotent, Rosselló et al. first showed that the cells could develop into primitive early embryos called embryoid bodies. For the vertebrate species tested, the embryoid bodies contained cells from each of the three main vertebrate embryo cell types. Secondly, iPSC-like cells from two organisms—chicks and zebrafish—formed various mature cell types when injected into developing chick or zebrafish embryos.

These results have two important implications. They suggest that the genetic mechanisms by which cells can be reprogrammed into a stem-like state have been conserved through 550 million years of evolution; additionally, they demonstrate that stem-like cells can be generated from important experimental organisms, and provide an important tool for both biological and biomedical research.

where it was once difficult to generate stem cells, such as in rats (*Li and Ding, 2010*) and pigs (*Wu et al., 2009*).

However, important issues in biology are addressed in experimental systems other than mammals, specifically in birds (*Jarvis, 2004, 2007*; *Jarvis et al., 2005*), fish (*Fetcho et al., 2008*), and flies (*Kuo et al., 2006*; *Yu et al., 2006*). Some of these animals have traits similar to humans that are not found in closely related non-human primates or commonly used laboratory animals. These include vocal learning in parrots and songbirds (*Jarvis, 2004*), widespread adult neurogenesis in non-mammalian vertebrates (*Nottebohm, 2002*; *Kaslin et al., 2008*), and vascularization and organ regeneration in zebrafish (*Poss et al., 2002*; *Stoletov and Klemke, 2008*; *Yaqoob and Schwerte, 2010*). Another important reason is that some traits are more easily studied in simpler organisms before they are applied to humans. The arthropod *Drosophila melanogaster* is an attractive genetic model due to the short life span, large number of offspring, and applicability of many genetic techniques (*van Ham et al., 2009*). *Drosophila* have been used to model Parkinson's, Huntington's, and Prion disease. Unfortunately, production of non-mammalian stem cells has been limited, due to problematic or unknown isolation procedures, and insufficient maintenance methods (*Lavial and Pain, 2010*). For these reasons, there has been a desire to generate stem cells for these species, allowing disease and mechanistic models to be explored, and possibly transgenic animals to be generated. Induced stem cells could provide such a model.

Here we attempted to generate an iPSC state for non-mammalian vertebrate and invertebrate model species spanning over 550 million years from a common ancestor (*Figure 1A*) (*Sullivan et al., 2006*): in birds (galliformes and songbirds), fish (zebrafish), and insect (*Drosophila*). We found that the four transcription factor genes used to induce mammalian stem cells can produce a partial iPSC state that varies with degree of relationship to mammals. Moreover, the mammalian (mouse) homolog of

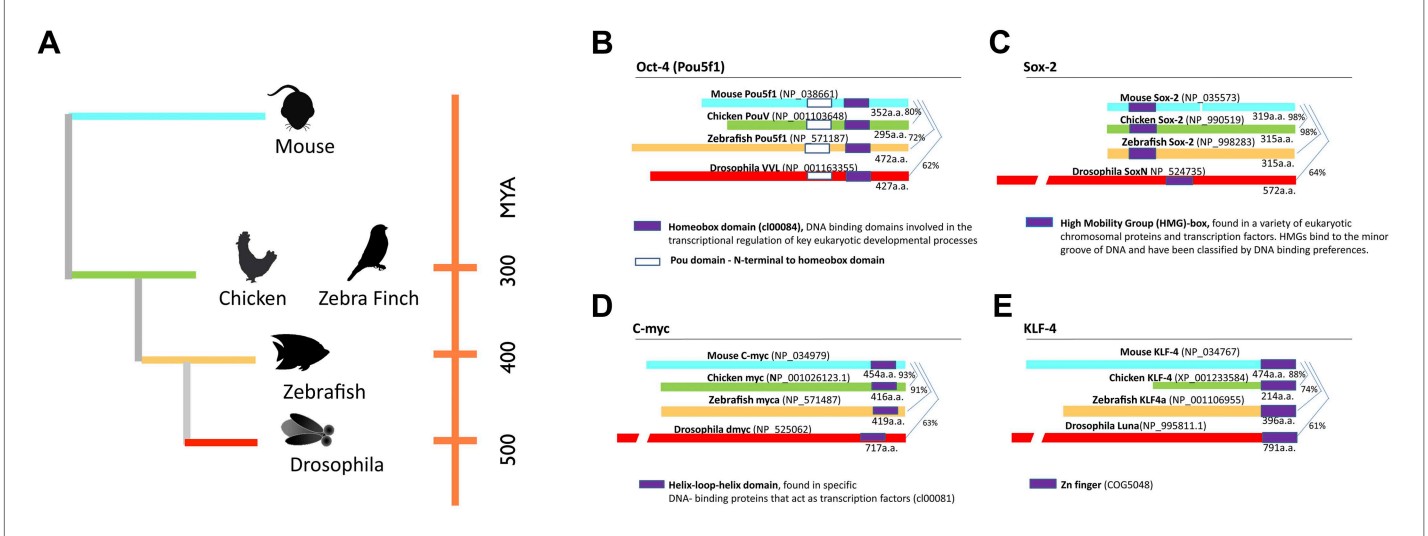

**Figure 1**. Phylogeny of species used and stem cell gene homologies. (**A**) Phylogenetic relationships of the species studied relative to mouse: birds (galliforms and songbirds), fish (zebrafish), and an insect (*Drosophila*). The phylogenetic tree is based on (***Sullivan et al., 2006***). (**B**) General structure and sequence comparisons of the putative homologs of the four stem cell inducing transcription factors included in the cassette (***Figure 1—figure supplement 1***; *Oct-4, Sox-2, C-myc, Klf-4*) across species. Although overall homologies vary significantly, DNA binding sites are highly conserved. Gene sequences were either from published studies (***Lavial et al., 2007***; ***Camp et al., 2009***) or from those predicted in sequence databases (Ensembl). Conserved domains (color coated boxed regions with accession numbers) were found using the Ensembl orthologue function and NCBI's HomoloGene. Detailed sequence homologies can be seen in ***Figure 1—figure supplement 1***.

The following figure supplements are available for figure 1:

**Figure supplement 1**. Schematic representation of the pHAGE-STEMCCA vector map.

**Figure supplement 2**. Alignments of the coding sequence of the putative Oct4, Sox2, Klf4, and c-myc homologs across vertebrate species.

**Figure supplement 3**. Alignments of the coding sequence of the putative Oct4, Sox2, Klf4, and c-myc homologs across invertebrate species.

these genes induced this partial iPSC reprogrammed state in the non-mammalian cells of all species tested, including inducing the ability of the vertebrate cells to incorporate into embryonic chimeras. We use the term partial iPSC or iPSC-like cells to denote cells that are transformed and show some iPSC characteristics. These findings are the first that we are aware of to generate iPSC-like cells across multiple non-mammalian species, using mammalian genes, in animal models where stem cells have been difficult or impossible to isolate (***Zwaka, 2008***; ***Lavial and Pain, 2010***).

## Results

### Induction of non-mammalian vertebrate cells with mammalian genes

In an ongoing effort to generate stem cells for transgenic songbirds with targeted gene manipulations, as opposed to non-targeted (***Agate et al., 2009***) as a means to study the molecular basis of vocal learning (***Jarvis, 2004***), we attempted the iPSC approach. We decided to transduce embryonic fibroblast cells of zebra finch and galliforms (quail and chicken) with a lentivirus retroviral vector (called STEMCCA [***Sommer et al., 2009***]) containing the four genes from the mouse driven by the human EF1α promoter (Map in ***Figure 1—figure supplement 1***). We surmised that the mouse genes might work in birds despite the separation of ~300 million years ago (MYA) from a common ancestor with mammals (***Figure 1A***), because although there were stretches of low homology and divergent sequences in three of the four genes (*Oct4*, *Klf4*, and c-*myc*), the overall conservation between birds and mammals was good (80–98% overall amino acid identity; ***Figure 1B–E***, ***Figure 1—figure supplement 2***). Furthermore, all four genes had highly conserved DNA binding domains (***Figure 1—figure supplement 2***; red boxes). For an iPSC positive control, we isolated mouse embryonic

fibroblasts and transfected them with the same lentiviral cassette (*Supplementary file 1A*). For non-iPSC positive controls, we used established ESC lines of mouse (*Nagy et al., 1993*) and chicken (*Pain et al., 1996*). For two negative controls, we transduced fibroblasts of each species with the same lentivirus vector, but containing GFP in place of the four mouse transcription factors, and grew the cells either in our stem cell media or complete media (see media composition in *Supplementary file 1B*). For a third negative control, we cultured non-transfected fibroblasts in stem cell media for each species to make sure media alone could not induce the cells (*Supplementary file 1A, B*). The two negative control groups grown in stem cell media exhibited similar qualitative and quantitative characteristics, and therefore, to diminish redundancy, the data shown is from the GFP-transduced fibroblasts. We repeated our experiments at least seven independent times, with 12–18 wells per species in 48 well plates ('Materials and methods'), and used established guidelines to evaluate iPSCs (*Maherali and Hochedlinger, 2008*; *Kim and Daley, 2009*).

The transformed avian cells showed a number of stem cell features absent from control fibroblasts and present in our mouse ESC and iPSC controls, and chicken ESC controls. This included, within 5 days, transformation from fibroblast morphology (*Figure 2A*) to colonies with characteristic clustered stem cell-like morphology (*Figure 2B*). These colonies had strong alkaline phosphatase (ALP) enzyme activity (*Figure 2D*), a characteristic of early and mature stem and tumor cells (*O'Connor et al., 2008*), whereas the starting fibroblasts did not (*Figure 2C*). They expressed Stem Cell Specific Antigen-1 (SSEA-1; *Figure 2F*), while none was detected in control fibroblasts (*Figure 2E*). An average of 20% of the wells had iPSC-like cells, as measured by colony morphology and ALP activity (measured from seven independent experiments for each avian species). Later iterations with different media conditions produced transformed cells in up to 90% of the wells (Dai et al., unpublished date). The higher the viral titer used, the more colonies were produced (*Figure 2—figure supplement 1*); the highest titer, $10^9$ U/ml, was used in the above experiments. We noticed some differences between the mouse and avian colonies, in that the mouse colonies as well as the individual cells within the colonies appeared on average slightly larger, while avian cells appeared more clustered. Similar differences have been observed when comparing human and mouse colonies (*Nichols and Smith, 2009*). The mouse and avian iPSC-like colonies were similar to those in established lines of mouse and chicken ESCs that we treated under the same growth conditions, including differences between the species (*Figure 2G–H*). These features were absent from our control mouse and avian fibroblasts treated under the same conditions with and without the lentiviral GFP-vector lacking the four transcription factors (*Figure 2A,C,E*; and not shown).

Like our mouse control iPSCs, the transformed avian cells (chicken, quail, and finch) expressed the four exogenous mammalian genes (*Figure 3A–D*; as determined by quantitative RT-PCR with mouse specific probes; *Supplementary file 1C*). After the first and second passages (3–4 weeks), three of the endogenous avian homologs (Oct4, Sox2, c-myc) were significantly upregulated 10–100-fold in the presence of their mammalian counterparts (except c-myc in quail; *Figure 3A–D*; green). The levels of induction of the endogenous and exogenous expression of these three genes in our chicken and mouse cells were similar to the control chicken and mouse ES cell. The level of induction in quail and zebra finch was lower (4–40-fold), but still statistically significant (p<0.0001, ANOVA) with no overlap in the expression detected in five replication experiments relative to the embryonic fibroblast controls. The fourth gene, Klf4, was upregulated in our mouse control iPSC and ESC, but not upregulated in any of the avian species (*Figure 3A–D*). However, *Klf4* was also not upregulated in the established control chicken ESC line (*Figure 3C–D*), relative to the chicken embryonic fibroblast. All avian species also showed significant induced expression of two other endogenous stem cell markers, nanog and vasa, not present in the STEMMCA vector, with levels more similar among species but lower than the mouse (*Figure 3E–G*). After about the fifth passage (2–3 months), the exogenous mouse genes were either completely (mouse and chicken) or partially (quail and finch) silenced, and this was associated with a concomitant further increase in some of the endogenous species-specific homologs (*Figure 3G–J*; including c-myc in quail as well as vasa and nanog, *Figure 3K–L*). However, Klf4 was still very low relative to the starting fibroblast controls in the avian cells, except for a small increase in some of the finch cell lines (*Figure 3J*).

Using modified media conditions containing differentiation inhibitors (Dai et al., unpublished date), we have been able to passage the iPSC-like chicken cells at the same rate as the mouse iPSC (currently > 20 passages) and these avian colonies still stain with ALP (*Figure 2—figure supplement 2* for the tenth passage) and the endogenous avian versions of the re-programming genes, with only minor differences compared to the fifth passage (*Figure 3—figure supplement 1* for the 12th passage).

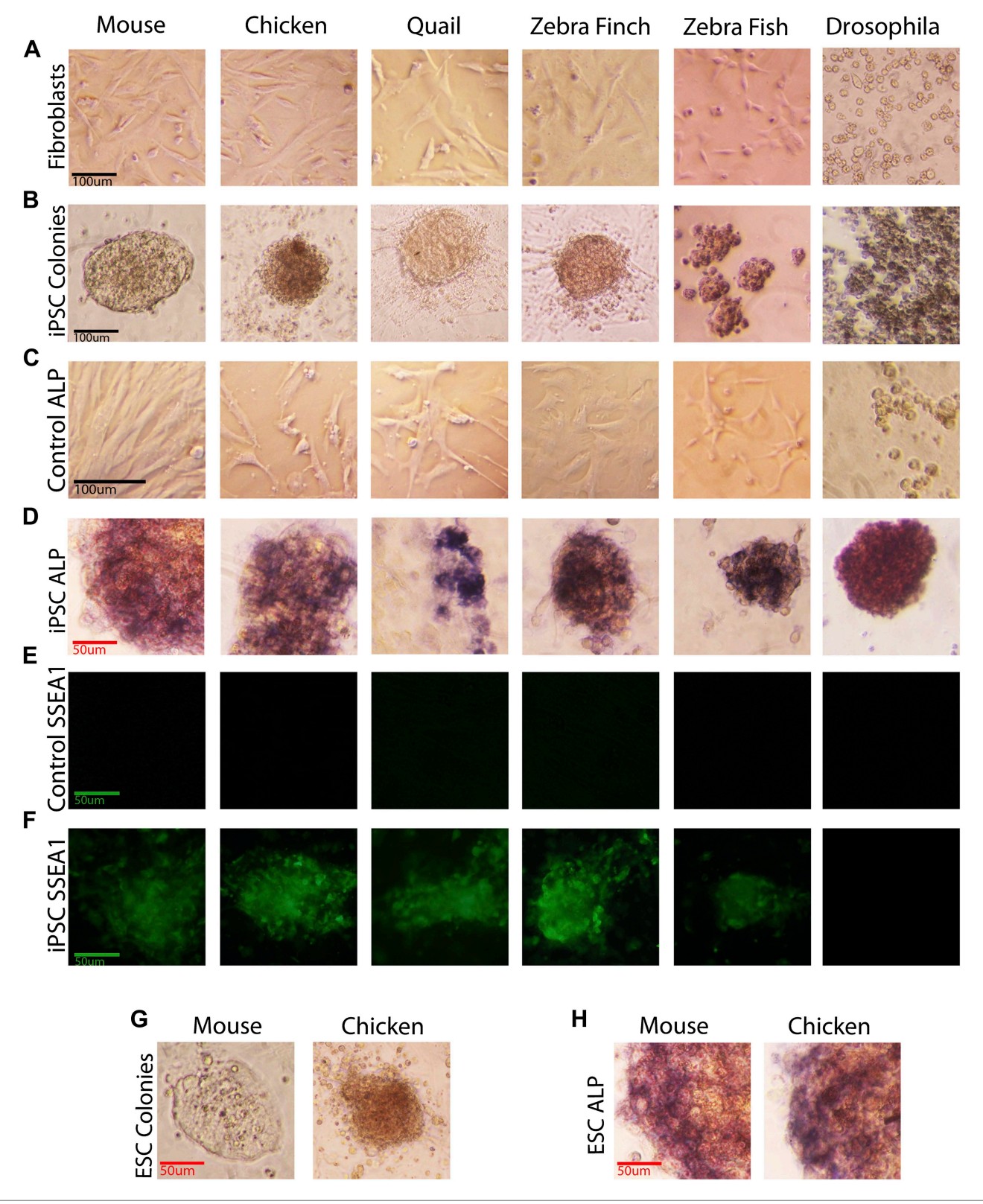

**Figure 2**. Generation of iPSC-like cells from differentiated cells of mouse, birds, fish, and *Drosophila* using the mouse transcription factors. (**A**) Non-transduced mouse, avian and zebrafish embryonic fibroblasts, and *Drosophila* S2 cell line. (**B**) Transformed cells (colonies) after 20 days (first passage), using optimal titers (***Figure 2—figure supplement 1***). (**C**) Non-transduced cells labeled for ALP activity. (**D**) Colonies formed by transformed
*Figure 2. Continued on next page*

*Figure 2. Continued*

cells labeled for ALP activity after the first passages (10th passage staining can be seen in ***Figure 2—figure supplement 2***). (**E**) Non-transduced cells and **F**, transduced cells after colony formation reacted with a Stage Specific Embryonic Antigen-1 (SSEA-1; green fluorescence) antibody. (**G**) Colonies of embryonic stem cells (positive controls). (**H**) Embryonic stem cells labeled for ALP activity (positive controls). Black scale bars, 100 µm; green and red bars, 50 µm.

The following figure supplements are available for figure 2:

**Figure supplement 1**. Colony formation in vertebrate cells as a function of species and titer.

**Figure supplement 2**. Alkaline phosphetase staining (red color labling) in chicken iPSC-like colonies after the 10th passage, and growth of fibroblast feeder layer cells that are not labeled.

When comparing expression of these genes in the iPSC cells with adult avian cells as opposed to the control embryonic fibroblasts, the relative levels of some factors (such as Oct-4) were still significantly increased above the adult levels (***Figure 3—Figure Supplement 2***). All of these findings were consistent for each avian species, given the low variation (S.E.M.) across independent replicates (***Figure 3A–L***, ***Supplementary file 1D***).

Based on this success, we mimicked transduction conditions for another non-mammalian vertebrate, the zebrafish (~400 MYA removed from mammals; ***Figure 1***), by transducing an embryonic clonal fibroblast line (ATCC, CRL-2147) with the STEMCCA lentivirus in fish-specific complete media supplemented with our stem cell media reagents (***Supplementary file 1B***). Although the homologies between mouse and fish for two (*Oct-4* and *Klf4*) of the four genes are less than they are with birds (***Figure 1B*** and ***Figure 1—Figure Supplement 2***), our rational to pursue this route was strengthened by a study that found that the downstream target genes of Oct4 are relatively conserved between zebrafish and mouse, and the mouse Oct4 can rescue zebrafish mutants (***Onichtchouk et al., 2010***). We found similar results for transformed zebrafish cells as for bird putative iPSC and ESC. This included cell colony formation (***Figure 2B***), ALP activity (***Figure 2D***), and expression of SSEA1 protein (***Figure 2E***), initial high expression (***Figure 3A–D***) and then silencing of the exogenous mouse genes by the fifth passage (***Figure 3G–J***), and absence of induction of endogenous *Klf4* (***Figure 3D–J***). There was also induction of the endogenous stem cell marker *Vasa* (***Figure 3E–F; K–L***). The only significant difference between the zebrafish and birds was lack of *Nanog* induction in the fish cells (***Figure 3E***). The average zebrafish colony size was also smaller (***Figure 2B***).

## Induction of an invertebrate cell with mammalian genes

Our results with vertebrate cells prompted us to consider whether these same mammalian genes can induce iPSC-like features in a yet more distant relative, in *Drosophila*, an invertebrate (550 MYA removed; ***Figure 1A***) (***Sullivan et al., 2006***). Although there are even greater divergences between mouse and *Drosophila* genes, we could still find putative homologs (62–64% identity) with highly conserved DNA binding domains (***Figure 1—figure supplement 3***). Thus, we transduced the commonly used *Drosophila* S2 line with the STEMCCA lentivirus or transfected with a plasmid containing the four factors and a Metallothionein inducible promoter. We decided to try both vector approaches, because, to our knowledge, there had been no successful attempts in transduction of genes into fly cells using lentivirus. Surprisingly, the lentivirus and its recombinant promoters worked in the *drosophila* cells, generating GFP labeled cells (***Figure 6—figure supplement 1C***).

We found that the transformed *Drosophila* S2 cells with the STEMCCA lentivirus or plasmid containing the four factors showed colony formation, although the colonies were notably fewer, smaller in size, and even darker than the vertebrate colonies (***Figure 2B***). The *Drosophila* colonies, like those of vertebrates, showed ALP activity (***Figure 2D***). They also expressed the exogenous mouse genes (***Figure 3M***) and, similar to the avian and fish transformed cells, the *Drosophila* transformed cells had induced expression of two of four endogenous homologues to the mammalian cassette, *VVL* (*Oct4* homolog) and *dMyc* (*c-myc*), low induction of *SoxN* (*Sox2*), and no induction of Luna (putative *Klf4*; ***Figure 3M***). There was also a significant upregulation of four of six other known endogenous *Drosophila* adult stem cell markers, *Dichaete*, *Escargot*, *Snail*, and *Vasa* (***Figure 3N***) (***Wilson and Dearden, 2008***; ***Palasz and Kaminski, 2009***).

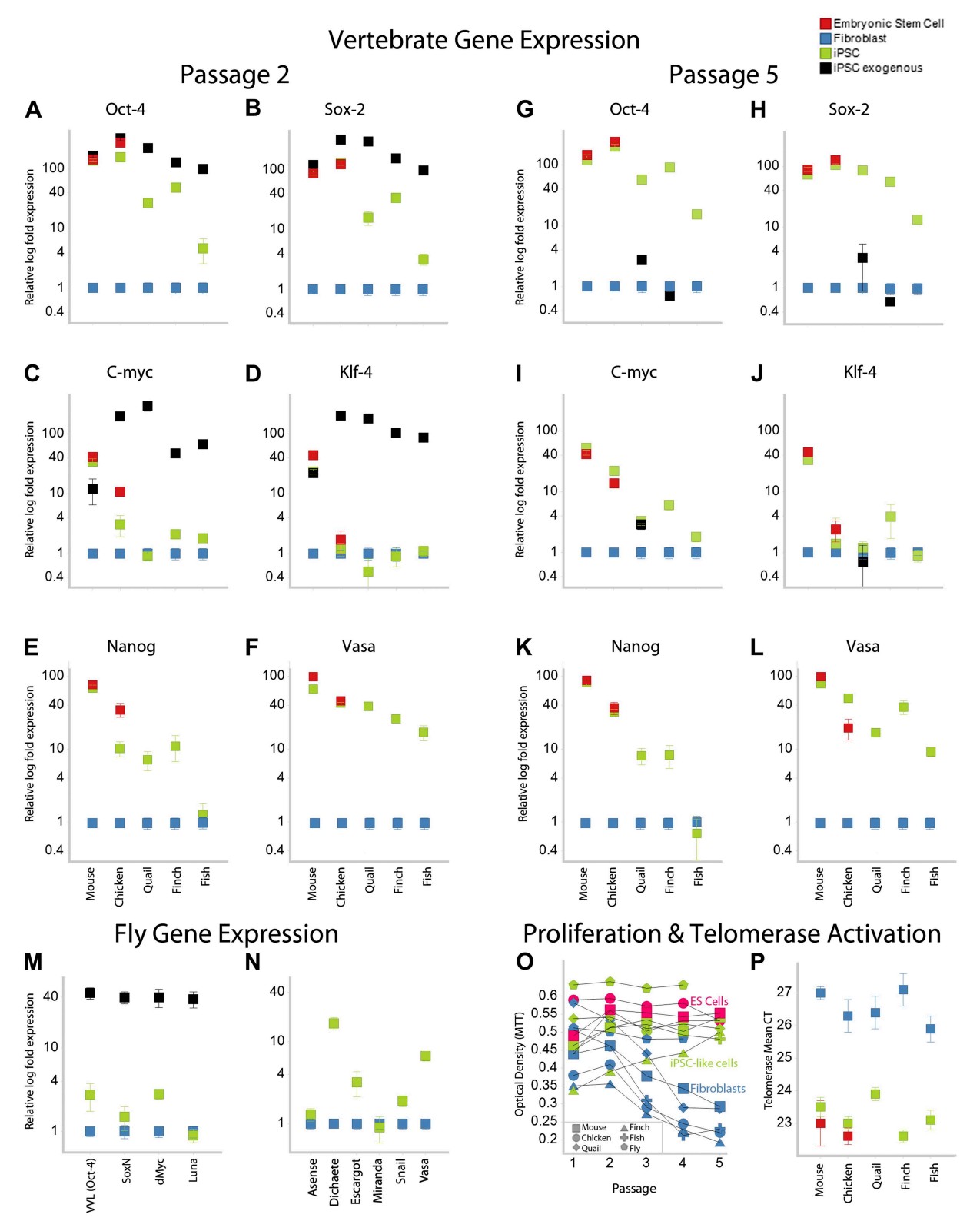

**Figure 3**. Upregulation of stem cell genes in mouse, birds, fish, and *Drosophila* by mouse transcription factors. (**A–D**) qRT-PCR of exogenous (black) mouse and endogenous (green) species-specific expression of *Oct-4* (**A**), *Sox-2* (**B**), *c-myc* (**C**), and *Klf-4* (**D**) in iPSC-like cells of each species after the second passage relative to (normalized) non-transduced fibroblast controls (blue). Mouse and chicken ESCs were included as positive controls (red).
*Figure 3. Continued on next page*

*Figure 3. Continued*

Primers used are shown in **Supplementary file 1C**. Several values overlap among cell types (e.g., mouse exogenous and endogenous *Oct-4* and *Klf-4*) and are thus not distinguishable in the graph. (**E**–**F**) qRT-PCR of *Nanog* (**E**) and *Vasa* (**F**) homologs in the different cell types across species. (**G**–**L**) qRT-PCR after the fifth passage show that the exogenous mouse genes are significantly downregulated or silenced. These values were normalized to the same fibroblast values as in the second passage. Nanog and Vasa expression levels exhibit no significant difference from passage two levels, except in chicken cells. Expression levels were also measured for 12th passage iPSC-like cells (**Figure 3—figure supplement 1**) and fifth passage iPSC-like cells were normalized against adult tissue (**Figure 3—figure supplement 2**). (**M**) qRT-PCR of exogenous and endogenous (homologs) *Drosophila* specific genes in the transformed S2 cells, and **N**, other genes known to be involved in early embryogenesis in *Drosophila*. Expression levels were also measured with iPSC-like cells generated from a primary *drosophila* cell line (BG2; **Figure 3—figure supplement 3**). Error bars, S.E.M within cell populations. p-values for all comparisons are shown in **Supplementary file 1D**, ANOVA, (Tukey's post hoc, p<0.001; n = 5 replicates of independent transformed lines). (**O**) Time course of self-renewal and proliferation of stem cells (iPSC-like cells and ESCs) relative to control fibroblast (or S2) as measured by the MTT [(3-(4,5-Dimethylthiazol-2-yl)-2,5-diphenilytetrazolium bromide] assay (read at 570 nm) (error bars not shown for clarity). ESCs and iPSC-like cells maintain high proliferation levels, while primary fibroblasts decay. (**P**) Telomerase activity was greatly increased (lower mean Cycle Threshold, CT) in iPSC-like cells and control ESCs over control fibroblast cells. Error bars, S.E.M (n = 5 independent cell line replicates for both MTT and telomerase data). Statistics shown in **Supplementary file 1D**.

The following figure supplements are available for figure 3:

**Figure supplement 1**. Comparison of iPSC-like expression patterns after the 5th passage and 12th passage.

**Figure supplement 2**. Gene expression profiles under different normalization basis.

**Figure supplement 3**. Drosophila BG2 cells also exhibited some transformation.

Because the starting S2 cells are polyploid and are known to be highly proliferative to begin with (**Moutinho-Pereira et al., 2010**), we wondered if these properties could have contributed to the induction process. Thus we tried another Drosophilia cell line, BG2, which is derived from the central nervous system and which is less proliferative (**Ui-Tei et al., 2000**). After 7 days and passaging, the BG2 cells also exhibited transformation phenotypes. Of nine independent transfection replicates in 48 well plates, three of them were successful, exhibiting morphological changes and expression of endogenous mouse genes (**Figure 3—figure supplement 3**), including in the majority (90%) of the wells. Thus the differences between experiments had more to do with other conditions than cell type differences between BG2 and S2 cells. However, there were differences between the two cell types (S2 and BG2) in the expression of induced genes. Like the S2 transformed cells, the BG2 transformed cells showed overexpression of SoxN, Escargot, Snail and Vasa. Unlike the S2 cells, the BG2 cells did not show significant overexpression of Diachaete, asense and VVL (**Figure 3—figure supplement 3**). This suggests that the starting state of the cells could make a difference, as seen with mammalian cells (**Kim et al., 2009**).

## Proliferation and telomerase activity

Proliferation levels of the mouse, avian, and fish transformed cells (measured by an MTT metabolic assay) were above their respective fibroblast controls after the first to third passage, depending on species (**Figure 3O**, green vs blue). The MTT levels at this time approached that of the mouse and chicken ESC lines (in red). Vertebrate cells with the GFP-vector alone treated under identical conditions instead showed a continuous decrease in MTT levels (in blue), which was clearly associated with senescence. The *Drosophila* result was somewhat different, since the control S2 cells are already highly proliferative. However, transformed S2 cells exhibited enhanced proliferation levels at all passages (**Figure 3O**; we did not assess the BG2 cells with MTT).

Despite the increased proliferation levels, we initially were not able to get the cells to grow well beyond the fourth to fifth passage. We thus attempted to determine optimal conditions for maintenance of the avian iPSC-like cells. We initially used chicken embryonic stem cell media (chicken ES media; **Supplementary file 1B**; [**Pain et al., 1996**]), which allowed transduction but resulted in few passages. We then discovered that with inclusion of 3i inhibitors of cell differentiation (**Li and Ding, 2010**) and doubling of LIF, growth was still slow, but the modified media supported proliferation up to about seventh passages (the condition used for most of our tests). Conversely, after decreasing LIF by half and doubling two of the 3i inhibitors, the transformed chicken iPSC-like cells became just as highly proliferative as the mouse iPSC and ESC cells (reported on in more detail in Dai et al., unpublished date).

We are currently passaging these chicken iPSC-like cells 1–2 times per week at a 1:4 dilution, and are above passage 20; we could be further along in passages, but froze the cells at various times over one year to postpone growth in order to conduct other experiments. Thawing the frozen cells does not prevent them from continuing to proliferate at a high rate.

Telomerase activity was also activated in all transformed vertebrate cells, and at levels comparable to those seen in the mouse and chicken ESC lines (*Figure 3P*). Telomerase activity is a characteristic feature of immortal cell lines (*Thomson et al., 1998*). Unlike the vertebrate cells, however, the transformed *Drosophila* cells did not have telomerase activity (not shown), confirming known absence of telomerase in the *Drosophila* genome (*Gomes et al., 2010*).

## Karyotyping

We karyotyped some of the avian species to assess chromosomal normalcy. The chicken iPSC-like lines (*Figure 4C*, male shown) displayed a normal karyotype of macro chromosomes in the majority of the spreads analyzed (90%; 18 out of 20) compared to a standard (*Nanda et al., 1999*), control fibroblasts (*Figure 4D*). The majority of the zebra finch (female) iPSC-like lines (90%) also displayed normal karyotype of macro chromosomes (*Figure 4E*, female shown). For zebra finch cells, standards were not available, and thus, the control fibroblasts were used as a reference (*Figure 4F*). The minority of cells that were not normal had tetraploid spreads, but in both iPSC-like and control cells: two out of 20 in the chicken iPSC-like cells and controls, two out of 20 in the finch iPSC-like cell, and one out of 20 in the finch control. This was a result of a doubling of the chromosome complement, which is common in cultured cells. These results with at least the avian cells suggest that major chromosomal arrangements did not occur as a result of the transformation.

## In vitro pluripotency

To assess pluripotency in vitro, we attempted to generate embryoid bodies (EB; 'Materials and methods' [*Takahashi and Yamanaka, 2006*]). Formation of EBs was achieved from the avian, fish, and *Drosophila* iPSC-like cells, and they appeared similar to those formed from our chicken and mouse ESC lines, and control mouse iPSCs (*Figure 4A*). The *Drosophila* EBs were more irregularly shaped. No EB formation occurred with the control cells of any of the species (fibroblast or S2), indicating that EB formation was specific to the iPSC-like cells and established ESCs. Differentiation into the three germ cell lineages was supported by quantitative RT-PCR of lineage-enriched genes showing over-expression relative to the fibroblasts of Brachyury (mesoderm), Nestin (endoderm), and Gata-4 (ectoderm) in all vertebrate species (*Figure 4B*) (*Leahy et al., 1999*; *Murakami et al., 2004*; *Hailesellasse Sene et al., 2007*). Conversely, the expression of these genes was much lower in our undifferentiated mouse, avian, or fish iPSC-like cells (i.e., the iPSC-like, green).

## In vivo pluripotency

The in vitro pluripotency results suggest that the iPSC-like cells have the potential to differentiate into multiple cell types, but EBs do not necessarily have advanced differentiated cell types, nor do they conclusively demonstrate the potential for incorporation in vivo. To assess pluripotency in vivo, we employed two strategies: generation of (1) teratomas and (2) chimeric embryos with the iPSC-like cells ('Materials and methods'). We did not attempt to do so with the *Drosophila* cells, as the early embryo is nearly one large cytoplasm partially divided up by membranes (*Mavrakis et al., 2009*). Teratomas were attempted for avian species by injecting the iPSC-like cells into the testes of SCID nu/nu mice in 18 animals for each avian species (nine with control fibroblasts and 9 with iPSC-like cells). After 35 days, two (out of nine) of the chicken iPSC-like and three (out of nine) quail iPSC-like cells injected mice developed teratomas. These teratomas exhibited organized formation of endoderm (such as neuronal rossetts, *Figure 5A,D*), mesoderm (such as bone, *Figure 5B,E*), and ectoderm (such as G.I Tract, *Figure 5C,F*), demonstrating pluripotency in vivo. None of the controls generated teratomas (*Figure 5H,I*). So far, none of the zebra finch iPSC-like cells formed teratomas, suggestive of possible species differences for in vivo pluripotency.

For the chimeric studies, we simultaneously transduced chicken and zebrafish fibroblast cells with the STEMMCA and the GFP lentiviruses, or transduced the cells with the GFP lentivirus after their second to fifth passage from frozen stocks. In both cases, we obtained GFP labeled colonies that still had the characteristic morphology of the iPSC-like cells (*Figure 6—figure supplement 1*). Cells were collected, washed, mechanically disassociated, counted, resuspended, and injected into embryonic 1-day (ED1) old chickens or 1–2 hr post fertilized (1–2hpf) zebrafish embryos, respectively. We then

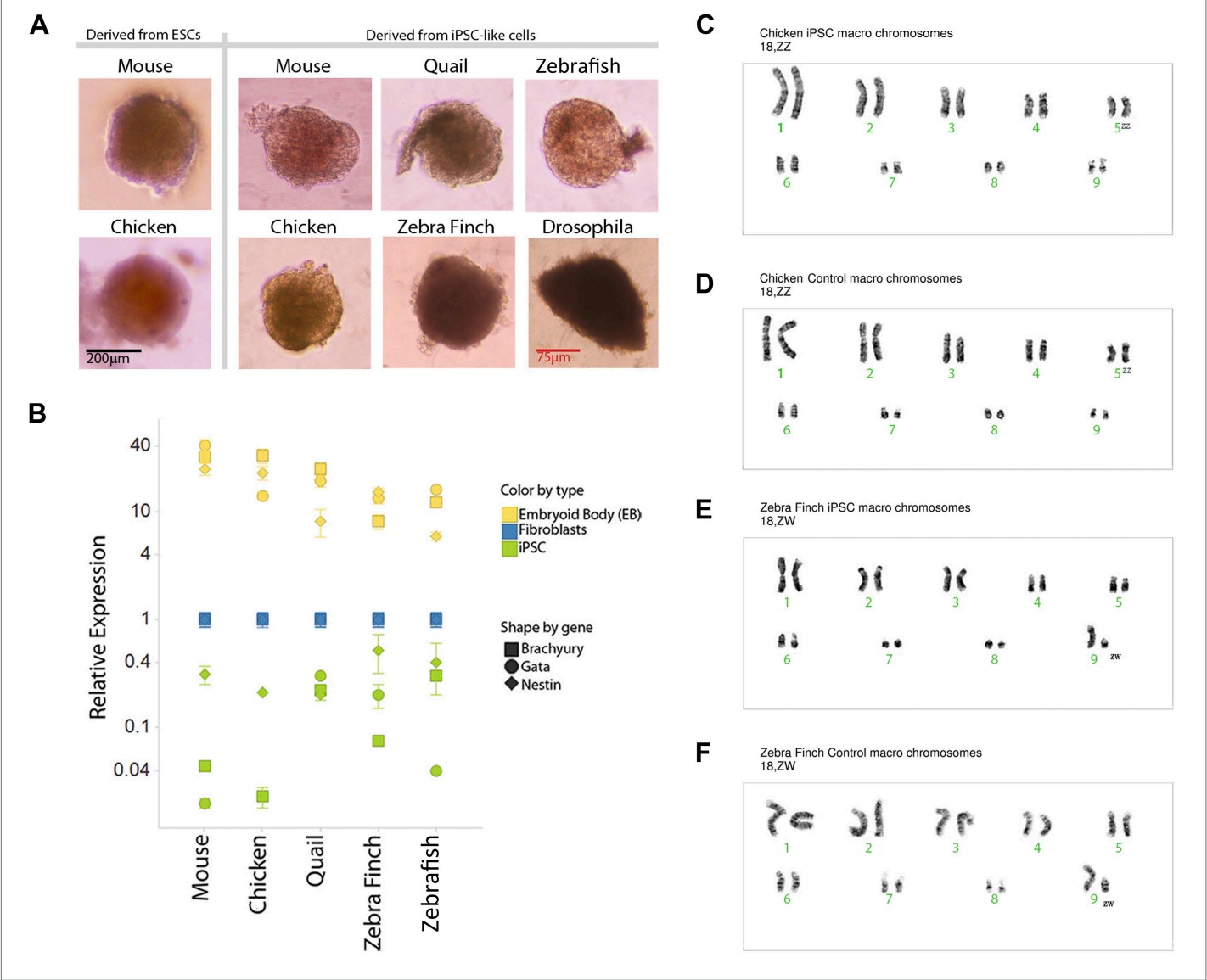

**Figure 4**. Karyotyping and in vitro pluripotency of iPSC-like cells. (**A**) Embryoid bodies (EB) from iPSC-like cells in differentiation media. (**B**) qRT-PCR gene expression analyses of Nestin (ectoderm marker), Brachyury (mesoderm), and Gata-4 (endoderm) homologs in undifferentiated iPSC-like cells (green) and in EBs (yellow) from mouse, bird and fish relative to their control fibroblasts (normalized; blue). Error bars, S.E.M. (n = 5 replicates of independently generated cell lines or EBs). Statistics in *Supplementary file 1D*. (**C–F**) Karyotypes of macro chromosomal arrangements of the chicken iPSC-like (**C**), chicken control fibroblasts (**D**), zebra finch iPSC-like cells (**E**) and zebra finch control fibroblasts (**F**), exhibiting 18 normal chromosomes. ZZ is female and ZW is male in birds. Black scale bar, 100 µm.

fixed the embryos 1–5 days later. We conducted control experiments in parallel with GFP-labeled fibroblasts (early first to second passage) injected into the embryos. We obtained animals up until ED4 for chicken and 72hpf for fish.

We found that recombinant GFP-labeled chicken and fish iPSC-like cells successfully incorporated into the developing animals (*Figure 5*). This required about 5000 cells for chicken and 100–200 for fish. The rate of chimera formation was about 16% for the chicken (four out of 25 attempts) and 10% for fish embryos (10 out of 103 attempts). Embryos injected with iPSC-like cells were subject to higher mortality than those injected with control fibroblasts. In chicken, about 50% of the embryos did not reach the 3rd day of incubation, compared to only 20% for control cells. Similarly with zebrafish, about 60% of embryos injected with the iPSC-like cells did not survive, while the rate was negligible

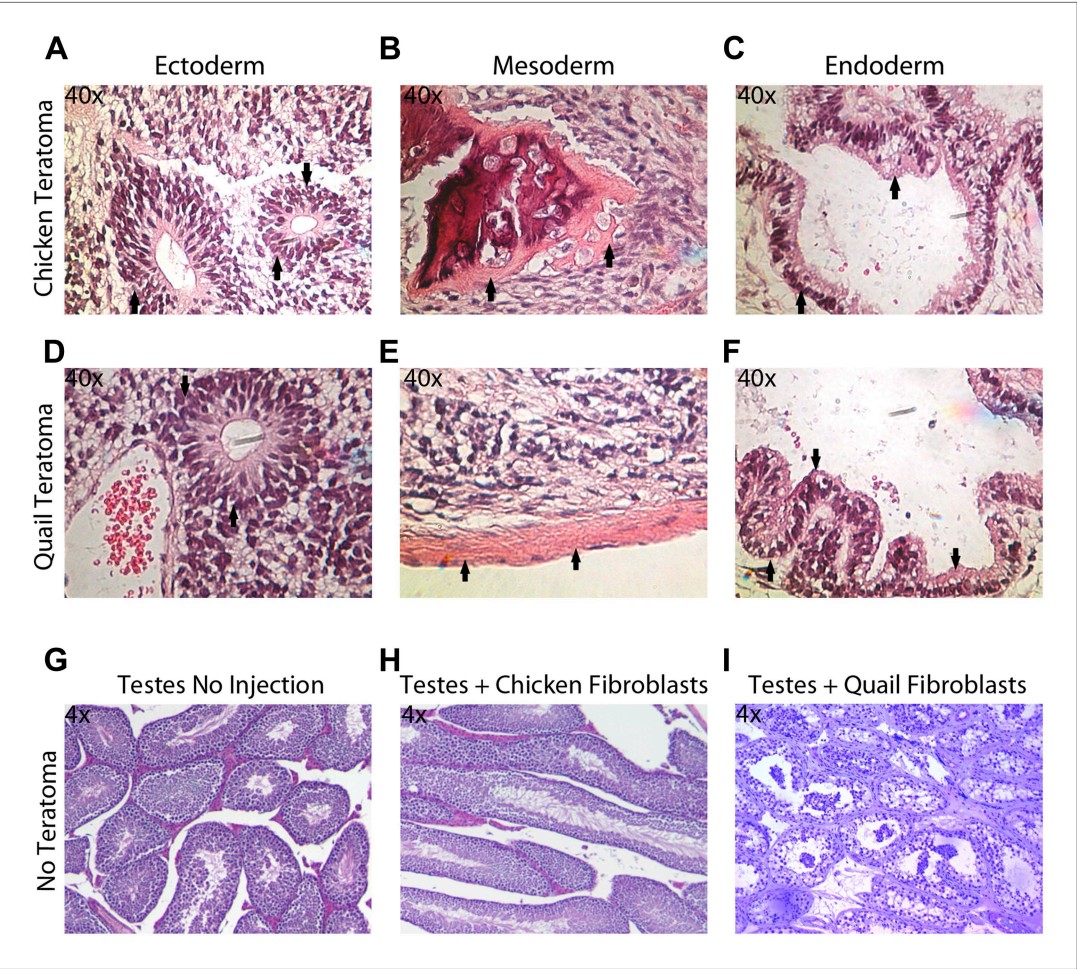

**Figure 5**. Teratoma formation generated by chicken and quail iPSC-like cells. (**A–C**) Teratoma formation after injections of chicken iPSC-like cells in testes of SCID mice, showing aberrant growth of (**A**) neural like cells (neuronal rosettes, endoderm, black arrows), (**B**) bone-like cells (mesoderm, black arrows), and (**C**) gastrointestinal tract-like cells (endoderm, black arrows). Similar features are seen in the quail generated teratomas (**D–F**). (**G**) Control testes without cell injections showing normal tissue morphology. (**H**) Testis with control chicken fibroblasts injected showing no germline formation. (**I**) Testis injected with control quail fibroblasts that did not generate teratomas. Panels **A–F** are at 40 × magnification, whereas **G–I** are at 4 × in order to get a broader view.

in control fibroblast injected embryos. These results are consistent with lower survival rates observed in iPSC-injected mice (***Maherali and Hochedlinger, 2008***), and could be due to multiple factors, such as the iPSC causing tumors and some other type of aberrant growth. Interestingly, after 1 day, some of the iPSC-like-GFP injected fish embryos produced a secondary axis (***Figure 6—figure supplement 2C***), suggesting a disruption in the developmental program. There were some zebrafish embryos, which, after 1 day of incorporation, exhibited a cluster of GFP labeled fibroblast derived cells (***Figure 6—figure supplement 2***), although to a lower intensity than the iPSC GFP homologs (***Figure 6—figure supplement 2B–C***). However, the starting fibroblast cells did not survive in the 72hpf fish embryos or the ED4 chicken, and thus did not generate fluorescently labeled older chimeras (***Figure 6A,E***).

Immunolabeling of GFP in tissue sections confirmed cell incorporation and allowed localization of the incorporated cells. In some animals (both chicken and fish), the cells incorporated in nearly all organs of the body (***Figure 6C,G***), but for the most part they incorporated sporadically (chicken iPSC-like cells incorporation, ***Figure 6—figure supplement 3***). In the chicken, the iPSC-like cells differentiated into many cell types, including into muscle, intestines, skin, and brain, while in the fish, most of the incorporation was observed in the stomach and the head (***Figure 6D***). A separate study will be conducted

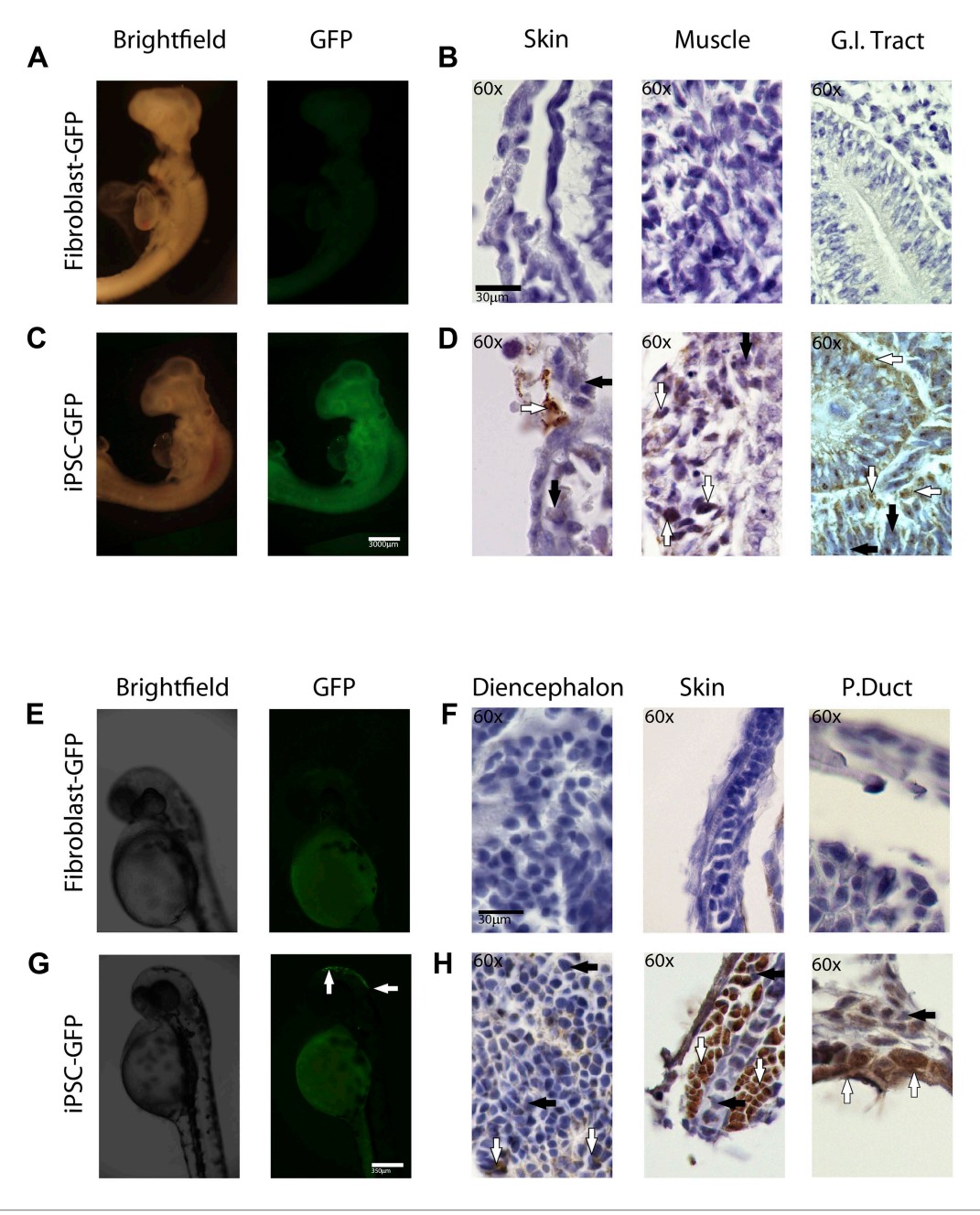

**Figure 6**. In-vivo pluripotency of iPSC-like cells from chicken and fish. (**A** and **C**) 4 day old chicken embryos that had been injected with GFP-labeled chicken fibroblasts (**A**) or GFP-labeled iPSC-like cells (**C**) 3 days earlier (GFP labeled cells in *Figure 6—figure supplement 1*). Incorporated GFP-labeled cells (green) are spread throughout the body for the iPSC cells but, not fibrobloast. (**B** and **D**) Histological sections stained with antibodies to GFP (brown) confirming absence of label in chicken fibroblast injected animals (**B**), and presence of label in multiple tissue types in the iPSC-like injected animals (**D**). (**E** and **G**) 3-day old (72hpf) zebrafish embryos injected with GFP-labeled zebrafish fibroblasts (**E**) or GFP-labelled iPSC-like cells (**G**), respectively. (**F** and **H**) Histological sections stained with antibodies to GFP (brown) confirming absence in controls (**F**) and presence of labeled cells in iPSC-like injected animals (**H**). Arrows in all images point to GFP-labeled cells; P.duct = pronephric duct (P. duct). 1 day old fish embryo is shown in *Figure 5—figure supplement 2*. Black bars, 30um; white scale bar, 3000 µm for the chicken and 350 µm for the fish. 1 day old post fertilization zebrafish embryos (*Figure 6—figure supplement 2*), and chicken embryos with partial incorporation (*Figure 6—figure supplement 3*).

*Figure 6. Continued on next page*

*Figure 6. Continued*

The following figure supplements are available for figure 6:

**Figure supplement 1**. iPSC-like cells for (A) chicken, (B) zebrafish, and (C) *Drosophila*, transfected with a GFP expressing lentivirus. Post induction to iPSC-like state.

**Figure supplement 2**. 1 day old post fertilization zebrafish embryos.

**Figure supplement 3**. Partial incorporation of chicken iPSC-like cells in chicken embryos as demonstrated by fluorescence and immunohistochemistry.

to see how long the embryos can live with the injected cells and whether they incorporated into mature gonads for germline transmission. No GFP immunolabel was detected in the 72hpf zebrafish or ED4 chicken controls (*Figure 6B,F*). These features are similar to those seen in the early mouse chimeras created with mouse iPSC cells (*Takahashi and Yamanaka, 2006*) and chicken chimeras created with chicken ESC cells (*Lavial et al., 2007*). Like those studies, most of our surviving embryos looked normal with no overt differences from animals treated under the same conditions without injected cells or injected with non-transduced fibroblast cells. These findings demonstrate that the non-mammalian vertebrate cells we generated with mammalian genes are pluripotent for at least non-germline cells in a developing animal in vivo, and functionally behave like mouse and chicken ESC and iPSC in vivo up until the ages analyzed.

## Discussion

Our results indicate that at least partially reprogrammed iPSC can be generated in non-mammalian species and that the mammalian genes are sufficient to do so in both non-mammalian vertebrates and invertebrates. While our study was under review, others (*Lu et al., 2012*) have recently shown that for one bird species, the quail, some of these properties can be induced using the human genes. We also had tested the four human transcription factors in all aves (chicken, quail, zebra finch) and zebrafish cells and found results similar to the ones obtained with the mouse factors ('Materials and methods' and data not shown). We cautiously call these cells partial iPSC or iPSC-like compared to authentic iPSCs, which refer to cells capable of giving rise to not only most cell types of an adult animal, but also to functional gametes for non-human species. Characteristics which the non-mammalian iPSC-like cells we generated have in common with mouse iPSCs and ESCs are colony morphology, marker expression of induced genes, reactivation of some endogenous pluripotency genes, transgene-independent self-renewal, embryoid body formation, teratoma generation (for chicken and quail), and the ability to contribute to different cell lineages in chimeric embryos (for chicken and zebrafish; *Table 1*). These findings suggest that the induction process is relatively conserved.

Some differences to mouse iPSC cells include lack or little induction of Klf4, overall initial slower growth of cells, lower overall fold-expression increase in endogenous stem cell genes, and presence of autologously derived fibroblast cells for some of the avian species in the middle passages (Dai et al., unpublished date); the latter two traits are similar to some human ES cells (*Draper et al., 2004*). For example, in the quail and zebra finch iPSC-like cells Oct4 is 20–80-fold higher than in control fibroblasts, but lower than the 100–400-fold increase seen in the chicken and mouse cells and yet not the 100–1000-fold increase typically seen in mouse cells (*Soldner et al., 2009*). We further note that our fibroblast controls were from early embryos, which already had some Oct4 and Nanog expression (detected in PCR reactions by the 28th cycle). Thus, we believe that not only protocol differences exist between studies that affect expression levels, but we clearly find species differences in induction levels for all four genes. Importantly, despite these species differences, the lower levels of one or more of these genes was still sufficient to generate pluripotent cells. Thus, we conclude that is it is not necessary for these genes to be induced 500–1000-fold in order for the cells to show some level of pluripotency in vitro or in vivo across species.

Apparent differences between species include SSEA1, Nanog, and Klf4 (*Table 1*). All species we studied showed induced *SSEA-1*, but this gene is not induced in human iPSCs or ESCs (*Takahashi et al., 2007*), indicating that human cells might be different from other vertebrates or at a different stem cell state. Nanog was induced in all species except zebrafish. Nanog is the third master transcription

**Table 1.** Comparison of characteristics of the IPSCs or PRPSCs cells across species

| iPSC or PRPSC cells | Stem Cell markers | | Induced endogenous homologs | Exogenous silencing | Gene expression | Self-renewal | | Pluripotency | | Chimera formation |
|---|---|---|---|---|---|---|---|---|---|---|
| | Morphology | Alkaline phosphatase | | | | Proliferation | Telomerase | EB formation/Teratoma | Gene expression | |
| Mouse | ESC-like clusters | YES | Oct4, Sox2, c-myc, Klf4 | YES | Nanog Vasa SSEA-1 | YES | YES | large aggregates | three germ lines | YES |
| Chicken | ESC-like clusters | YES | Oct4, Sox2, c-myc, **−Klf4** | YES | Nanog Vasa SSEA-1 | YES | YES | large aggregates/Teratomas formed | three germ lines | **YES** |
| Quail | ESC-like clusters | YES | Oct4, Sox2, c-myc, **−Klf4** | YES | Nanog Vasa SSEA-1 | YES | YES | large aggregates / Teratomas formed | three germ lines | ND |
| Finch | ESC-like clusters | YES | Oct4, Sox2, c-myc, **−Klf4** | YES | Nanog Vasa SSEA-1 | YES | YES | small aggregates | three germ lines | ND |
| Zebrafish | ESC-like clusters | YES | Oct4, Sox2, c-myc, **−Klf4** | YES | - ***Nanog*** Vasa SSEA-1 | YES | YES | small aggregates | three germ lines | **YES** |
| *Drosophila* | **Darker, some clusters** | YES | Oct4 (WL), SoxN, d-myc, **−Luna (klf4)** | ND | Vasa Dichaete Escargot Snail | YES | NA | small aggregates | NA | ND |

A large number of similarities are found. Species differences are highlighted in bold. NA, not applicable; ND, not done.

factor in the stem cell regulatory system (*Nichols et al., 2009*; *Silva et al., 2009*) that promotes self renewal in the absence of LIF (*Chambers and Tomlinson, 2009*). However, recent studies suggest that Nanog may not be integrated in a pluripotency regulatory circuit in fish (*Camp et al., 2009*). We note, though, that the so-called fish Nanog homolog (NCBI Accession # NP_001091862) of these and many other studies show low sequence identity across much of the protein coding sequence with birds (32%) and mammals (31%). In this regard, it is possible that the fish Nanog homolog has not been really identified or is not present in the zebrafish genome. The absence or very low induction of endogenous *Klf4* in all of the non-mammalian species may either be due to an inability of the mammalian genes to re-program this gene, redundancy of the Klf family (*Jiang et al., 2008*), or a lineage specific difference of mammals. The later hypothesis is supported by the low expression of *Klf4* in chicken ESCs relative to control fibroblasts. Induction of endogenous *Klf4* has so far been found in several mammalian species, such as pig, mouse, and rats (*Roberts et al., 2009*). It has been shown that Klf4 preferentially regulates genes involved in cell adhesion, either activating or inhibiting adhesion, and that cell adhesion can inhibit proliferation (*Swamynathan et al., 2008*). This function is consistent with our findings that relative to mice, the non-mammalian iPSC-like cells are more adhesive to each other, which is known to slow proliferation even in mouse iPSCs (*Chen et al., 2011*). Future investigation could test these hypotheses by over-expressing mouse or species-specific Klf4 without silencing, and assessing long-term proliferation.

The induced expression of endogenous Oct4, Sox2, (and partially of c-myc) homologs in all species and their continued expression when the mouse transgenes became silenced suggests that these genes may be playing more conserved inductive roles. Oct-4 and Sox-2, along with Nanog, are known as master transcription factors for the pluripotent stem cell state in mammalian cells (*Chan et al., 2011*). Oct4 and Sox2 can dimerize and when bound to DNA motifs of their target genes, they activate gene regulatory networks involved in both self-renewal and pluripotency (*Remenyi et al., 2003*; *Shi and Jin, 2010*). Sox2 alone has been shown to play various roles in different tissue types or cell states (*Tomioka et al., 2002*). Oct-4 has been deemed the most important of these master factors in the mammalian stem cell regulatory system (*Sterneckert et al., 2011*). Although not a master factor, c-myc is known to induce proliferation, by repressing growth arresting genes (*Gartel and Shchors, 2003*). This makes it a key contributor in inducing the self-renewal state of the cell. Recently, other factors that are less oncogenic have been shown to be suitable substitutes for c-myc, such as Gliss1 (*Maekawa et al., 2011*). These substitutes may be useful in future studies in non-mammalian species. A recent study also showed that an intermediate state of stem cell induction can exist, by using just a few of the transcription factors (*Lin et al., 2011*). Different from our iPSC-like cells they did not successfully achieve pluripotency in vivo. Thus, our cells appear to, at least, be further along in the programming stage than these cells.

All together, we show that there are at least 7–8 stem cell marker genes up-regulated and three lineage specific germ layer genes down-regulated in the induced cells across species, relative to the embryonic fibroblasts. While we have not performed full-scale transcriptome analysis or DNA methylation studies, the results suggest that enough of a gene regulatory network was induced for the vertebrate cells to maintain a stem cell-like state in the absence of continued exogenous mouse transgene intervention (*Maherali and Hochedlinger, 2008*), to become stably proliferative in optimal media conditions for the avian cells, and to become pluripotent in vivo.

It is important to caution several factors about the *Drosophila* results. First, the induction of stem cell characteristics was less prominent than in the vertebrates. When embryoid bodies were generated, they did so at a much lower rate with *Drosophila* cells than with the vertebrate cells. Putative endogenous homolog expression did not occur in two of the genes for the S2 cells and in another three for the BG2 cells. Second, proliferation was enhanced, but the starting S2 cells were already relatively proliferative. This difference might be because S2 cells are partially differentiated, aneuploid, renewable cells (*Moutinho-Pereira et al., 2010*). As such, these cells are a work-horse type of cell line, such as HeLa cells for humans, more so than true primary somatic cells. One interpretation of these findings is that the S2 cells underwent transdifferentiation as a result of the presence of mammalian stem cell genes, but for a stem cell state; that is switching from one highly proliferative state to another. Altogether, the findings suggest that the closer the relationship to mammals, the more reprogrammed characteristics the cells showed.

Although we were hoping that the mammalian genes would induce stem cell-like cells in other species, we were quite surprised that they did so and at the efficiency discovered. Substituted gene

function among vertebrates (*Enard et al., 2009*) and between vertebrates and invertebrates (*Lavial et al., 2007*) has been demonstrated previously, but we are not aware of a systematic set of genes doing so. It is possible that a stem cell gene regulatory network and the stem cells themselves share more conserved molecular similarities than differentiated divergent cells. This idea is supported by the fact that Oct4 has been shown to regulate some of the same genes, including Sox2, in mammals and fish (*Onichtchouk et al., 2010*), and embryonic cells and the three germ layers are more similar to each other in distantly related organisms than their adult cells, which are more divergent (*Parikh et al., 2010*). This is further supported by the finding that the iPSC features were maintained in the vertebrate cells even after the mouse transgenes were suppressed. As such, our findings suggest that stem cells could represent a primordial animal cell state conserved across the animal kingdom. This can be tested by comparing the transcriptomes of iPSCs and more differentiated cell types across species.

Future studies may also focus on transducing cells with their species-specific genes, testing non-integrating genome mechanisms of transduction, using promoters that have been shown to work at high efficiencies in a particular species, super-induction and inducing cells to different stem cell states (*Ye and Cheng, 2010*). Along these lines, it could even be possible for different stem cell types, such as germ cells, epi-stem cells, and other primed states to be present within the current conditions (*Dejosez and Zwaka, 2012*).

In summary, the generation of in vivo incorporating stem cells for non-mammalian species should help advance studies of these experimental model systems. These cells can be used as tools to study differentiation, evolution, and disease across a wide range of species, including cancer (*Takashima et al., 2013*). The induced cells might serve as a platform to study cellular evolution at the molecular level.

## Materials and methods

### General cell culture

Embryonic fibroblasts were collected at embryonic day 12.5 for mouse and the comparable stages (embryonic day 8) (*Butler and Juurlink, 1987*) for chicken, quail, and zebra finch. Briefly, several embryos (n = 4) were extracted from the womb or eggs, their head, limbs, and liver removed, and the remaining contents were minced manually using forceps. The minced contents were placed in a 15 ml tube and treated with 0.25% trypsin (0.25% Trypsin/EDTA, Gibco; 1–2 ml per embryo) for 30 min at 37°C, pipetting briefly every 5 min to enhance dissociation. Trypsin was neutralized with complete media (*Supplementary file 1B*), cells were spun down, counted (hemocytometer), re-suspended in complete media and plated at a concentration of one embryo per 150 mm dish for mouse and per 100 mm or 60 mm dish for chicken/quail and zebra finch, respectively. When grown to confluent layers, all fibroblasts were passaged in complete media twice before cells were frozen in aliquots. Zebrafish fibroblast cells were purchased (ATCC, CRL2) and maintained per supplier's specifications at 26°C in zebrafish complete media (*Supplementary file 1B*). For *Drosophila*, Schindler's cell line (S2), an epithelial-like cell line, was purchased (ATCC, CRL 1963) and passaged (1:10) and maintained per supplier's specifications in *Drosophila* complete medium (*Supplementary file 1B*). BG2 cells were purchased from the *Drosophila* Genome Research Center (ML-dmBG2; number 51), and maintained with growth culture conditions provided by the center. Mouse embryonic stem cells (ESCs; line R1 [*Nagy et al., 1993*]) were cultured using standard conditions (*Joyner, 1999*). Chicken ESCs (25th passage) were provided by Dr Bertrand Pain (Clermont University, France) and cultured according to their protocol (*Lavial et al., 2007*). Adult cell lines for mouse, aves, and fish were either generated or purchased (*Supplementary file 1E*).

### Vectors

Lentiviral vectors were generated in human embryonic kidney (HEK) 293T cells (Cell Biolabs, San Diego, CA, Cat # LTV-100), using a third-generation lentiviral system, following a previously described protocol (*Cockrell and Kafri, 2007*). Prior to transfection, the cells were plated on 10 cm collagen coated plates at a density that resulted in 60–70% confluency at the time of transfection. A transfection mix was prepared with either 5, 10, or 15 µg of DNA of the STEMCCA vector or control GFP lentiviral vectors (EF1α-GFP; both kindly provided by Dr Gustavo Mostoslavsky), packaging cassette (REV and Gag/Pol, 10 µg) and the VSV-G (5µg) envelope expression cassette, respectively. The cells were then transfected with the mix, using 40 µl of Lipofectamine (Invitrogen, Carlsbad, CA) per plate. 8 hr after the addition of DNA, the transfected cells were washed with PBS and fresh complete media as used for mouse

cells (*Supplementary file 1B*). Media with viral particles were collected every 24 hr for the next 48 hr and stored at 4°C until complete. Viral particles were separated from cellular debris by centrifugation at 4000g for 5 min followed by filtration through a 0.45-micron filter. The titer was measured using Quick-Titer (Cell Biolabs Inc, Cat # VPK-112) and promptly stored at −80°C. If necessary, titer concentrations were increased by ultracentrifugation (SW-29 rotor) at 50,000g for 2 hr, followed by re-suspension in PBS (pH = 7.2).

We also used a commercially available human stem cell cassette with GFP (Biosettia, cat# iPSC-p4F01) on the avian cells. We established DNA preps and lentiviral vectors as above. Maximum titer was significantly less than with the STEMCCA cassette ($2.5 \times 10^8$ U/ml). For *Drosophila* transductions, we also generated a plasmid with the Metallothionein inducible promoter from the vector pMT/BiP/V5-His A (Invitrogen). The four transcription factors in the STEMMCA cassette described above were cloned into pMT/BiP/V5-His A in two steps: first, the Oct-4 and Klf-4 segment, followed by the Sox-2, c-myc segment. The cloning was confirmed by sequencing using plasmid and gene specific primers.

## Transduction of cells and iPSC culture

Transduction was performed using the ViraDuctin system, as per supplier's protocol (Cell Biolabs, Cat # LTV-201) in complete media (*Supplementary file 1B*). Before transduction, cells were thawed and cultured in complete media until 80% confluent. After transduction, cells were grown for 5 days (2 days for *Drosophila*), then passaged (first passage), and let to grow for approximately 20 days (8 days for *Drosophila*) in 3i Media (Stem Cell Sciences, UK, SCS-SF-ES-01) or our mouse stem cell media for mouse cells (*Ying et al., 2008*), our modified version of avian stem cell media (*Pain et al., 1996*) for avian cells, fish stem cell media and *Drosophila* stem cell media (*Supplementary file 1B*). *Drosophila* cells grew faster than the vertebrate fibroblasts, and thus, markers were observable at earlier time points. Cells of all species were then subsequently passaged when cultures reached confluency, which was every 7–10 days for the vertebrate cells or every 3 days for *Drosophila* cells, and divided 1:2 (Aves and Fish) or 1:10 (*Drosophila*, due to more rapid growth). Before we performed detailed analyses on multiple transfections, viral transduction efficiency values were assessed at three different STEMCCA concentrations in 48 well plates and cell colony forming units quantified in the vertebrate species (*Figure 2—figure supplement 1*). We measured 11 independently transduced plates, and analyzed differences based on titer and species. Based on these initial transduction experiments, most subsequent transductions were performed at $10^8$ U/ml for mouse and $9.5 \times 10^9$ U/ml for all other species (birds, fish, fly) to achieve similar colony forming unit levels as starting points for our analyses. For subsequent analysis, in order to achieve statistical confidence, we transfected 12 to 30 wells seeded with primary cells, in seven different independent experiments. Each well was independently transfected. Samples of the cells were then extracted at various time points, to identify the presence of exogenous or endogenous genes and proteins, via RT-PCR and immunocytochemistry, respectively. For all species, negative control groups were conducted utilizing fibroblasts transduced with a GFP containing lentivirus and grown in the stem cell media (*Supplementary file 1A*). For in vivo pluripotency experiments, both fibroblasts and iPSC-like cells were first transduced with the GFP lentiviral vector (titer $10^8$), following the same transfection protocol. We also performed post induction GFP transfection on the *Drosophila* cells, although these were not used for in vivo studies.

To transduce S2 cells with the Metallothionein inducible promoter plasmid, we used a previously described protocol (*Santos et al., 2007*). To induce expression of the transcription factors, 1–2 days after transfection, copper sulfate was added to the medium to a final concentration of 500 µM (5 µl of a 100 mM CuSO4 stock). To transduce avian cells with the human STEMCCA lentivirus, we used the protocol described in the preceding paragaph. Colonies were observed after around the 7th day, but they numbered less than with cells transduced with the mouse genes. These colonies showed alkaline phosphatase staining and formed embryoid bodies (not shown).

## qRT-PCR

Cells or embryoid bodies were spun down and RNA isolated using a standard kit (Promega SV total RNA isolation system, Z3105). RNA was quantified using a NanoDrop 2000c (Thermo Scientific, Waltham, MA) and then stored in −80°C. Complementary DNA (cDNA) was produced by reverse transcription (RT) in a 20 µl reaction using the supplier's protocol (10 µl of 2X RT buffer and 1 µl of 20X Superscript II enzyme; Applied Biosystems). The cDNA was then used as a template to perform PCR gene expression assays in 20 µl reactions containing 1 µl template (~2 µg/µl), 10 µl 2X Gene Expression Master Mix (BioRad, Hercules, CA) and forward and reverse TaqMan primer probes (Generated by Applied Biosystems)

or in 20 µl reactions containing the same reagents, but in place of TaqMan primers, custom PCR primers and 1 µl SYBR green (BioRad). To discriminate between endogenous and exogenous expression of the stem cell genes across species, different primers were generated for mouse and the non-mammalian species, using non-overlapping sequences. To discriminate between mouse exogenous and endogenous genes, primers to the WPRE region of the vector were used. Using this strategy, the estimated relative amount of endogenous expression was calculated as the expression level of the WPRE segment subtracted from the total RNA of the mouse specific transcription factors. Primer sequences are listed in **Supplementary file 1C**. The reactions were performed in a Cx96 real-time machine (BioRad). Cycling conditions were 95°C for 10 min, followed by 35 cycles of denaturation at 95°C for 15 s and annealing/extension at 60°C for 1 min. No-template controls were run for each primer set and probe. 18S rRNA endogenous control was run for each sample using TaqMan primers that recognized the RNA in all species tested (Cat# Eukaryotic 18S RNA HS99999901_S1; Applied Biosystems). The results were normalized to the endogenous 18S expression and to the gene expression level of the control fibroblast/primary cell groups using the ΔΔCT method common for RT-PCR analyses. All primers showed efficiency levels above 90%, using the protocol in the MIQE guidelines (minimal information for publication of real-time PCR experiments) (**Bustin et al., 2009**). For statistical analysis, 2-way ANOVAs were performed on two factors (genes and cell types [iPSC, fibroblast, ESC, EB]) on n = 5 independently transduced lines (replicates) for each of the vertebrate species or n = 3 independent lines for the *Drosophila* cells.

## Alkaline phosphatase

Alkaline phosphatase (ALP) activity was measured using the STEMTAG Immunohistochemical Kit (Cat# CBA-300, Cell Biolabs), following the manufacturer's protocol. Control fibroblasts, ESCs, and iPSCs were washed with PBS, and fixed with either 4% paraformaldehyde or the kit's fixing solution for 10 min at room temperature. The fixing solution was then aspirated, the staining solution was placed in each well for 30 min and stored in the dark at room temperature. The wells were washed with dH$_2$0 3 times and images were taken immediately under a stereomicroscope without coverslipping. A dark blue/purple color product indicates the presence of ALP enzymatic activity normally found in stem cells, whereas differentiated cells will not stain. The same protocol was also employed, in some instances, with Vector Red as an indicator (Vector Laboratories, inc, Burlingame, CA).

## MTT (proliferation) assay

To assess proliferation, we used the MTT (3-[4,5-Dimethylthiazolyl-2]-2,5-diphenyltetrazolium bromide) Quantitative Cell Proliferation Assay (Cat# 30-1010K; ATCC). Tetrazolium salts are reduced metabolically by the cells, resulting in a colorimetric change. The resulting intracellular purple formazan is solubilized and quantified spectrophotometrically (at 570 nm). Cells (induced and controls) for all species were plated at 10,000 cells/well (in quintuplets, from independently transduced cells) and incubated for 24 hr. 10 µl of the MTT reaction solution was added to each plate and incubated for 3 hr. 100 µl of detergent was added to each plate, stored for 2 hr in the dark (room temperature), and the absorbance was measured at 570 nm using a Molecular Devices Emax Microplate Reader. ANOVA was performed to test for differences between cell types and species (n = 5 independent lines, per species). Statistical significance was considered at p<0.05.

## Telomerase activity

Telomerase expression is low or absent in most somatic tissues, but not in germ cells, stem cells, and tumors (**Meyne et al., 1989**). The telomerase binds to a particular repeat seq,uence TTAGGG present at the ends of chromosomes of most eukaryotic species and extends them during cell replication. Telomerase enzymatic activity was determined using the Quantitative Telomerase Detection Kit (BioMax, USA, MT3012), following the manufacturer's protocol. Cell extracts containing proteins and RNA were generated from the ESC, iPSC, and control fibroblast, and then telomerase activity was measured. If telomerase is present, it adds nucleotide repeats to the end of an oligonucleotide substrate of the kit, which is subsequently amplified by real time qPCR. Quantitation was carried out by the PCR software of the BioRad Cx96 system. Positive control (template provided with kit) and negative control (heat inactivated samples) reactions were performed. Cycling conditions for the BioRad Cx96 real-time machine were as follows: 48°C for 10 min and 95°C for 10 min, followed by 40 cycles of 95°C for 15 s (denaturation) and 60°C for 1 min (annealing/extension). All reactions were

performed in quintuplets. Paired *t*-tests were performed to test for differences of telomerase in the iPSC-like and control fibroblasts of each cell line. Statistical significance was considered at p<0.05.

## Karyotyping

Karyotyping was performed as previously described (*Bangs and Donlon, 2005*), by Karyologic, inc. Briefly, cells were seeded in t-25 tissue culture flasks, and allowed to grow. Colchicine (Colcemid, Invitrogen 15210-040) was added to each flask (0.25 ml/5 ml media) and incubated at 37°C, 5% CO2 for 12 hr. Cells were then trypsinized, transferred to 15 ml tubes and spun down at 1200 RPM, for 8 min. Cells were then resuspended in 0.0075 KCL and incubated at room temperature (6 min) before being spun down again. Cells were then fixed with Methanol/Acetic acid fixative (3:1) and stored overnight. Cell suspensions were then dropped into cold slides, dried and baked for 20 hr at 65°C. In order to assess the banding of the chromosomes, slides were treated with 0.05% trypsin 0.02 EDTA at room temperature for 12 s, rinsed quickly in 100% ethanol and then in Gurr's phosphate buffer (pH 6.8, Invitrogen #10582-013). Slides were then stained with Karyomax Giemsa (Invitrogen #10092-013), per manufacturer instructions. To assess the chromosomes, Applied Imaging Genus Cytovision Software (v2.8) was used.

## Embryoid body formation

In order to form embryoid bodies (EBs), the hanging drop method was used (*Keller, 1995*). After harvesting the iPSCs or control fibroblasts (or S2 cells) directly from culture on the stem cell media, they were washed with PBS (pH7.4; Gibco) to remove any LIF and resuspended in 'differentiation media' which is complete media for each species excluding LIF, cytokines, chemical inhibitors and mercaptoethanol. The cells were then micropipetted in 20 µl drops containing ~500 cells each on the lids of bacteriological plates (Sigma, 100 mm). The lids were inverted over a dish filled with 10 ml PBS and incubated for 2–3 days. After the embryoid bodies had formed from the iPSC-like cells, the drops were flushed from the lid with differentiation media and grown in suspension culture for another 3–5 days. Embryoid bodies were then collected via pipette, RNA extracted (as above), and qRT-PCR analysis conducted (as above).

## Immunohistochemistry

For SSEA-1 labeling, reactions were performed on cells cultured on coverslips in 24 well plates. The primary SSEA-1 antibody (Cat# 480, Santa Cruz Biotechnology, Dallas, TX) was diluted (1:200) in PBS. A secondary anti-mouse IgM conjugated to a green fluorescent molecule (Abcam, Cambridge, MA) was diluted (1:500) and incubated at 4°C, overnight. The cells were then washed 3X in PBS and coverslipped with DAPI solution (VectaShield; Vector Labs). Images were taken using a fluorescent microscope (Olympus Bx61).

For GFP labeling (performed by the Duke University Pathology Lab), chicken or zebrafish embryos, or positive control tissue slides (Mouse GFP positive brain sections), were cut at 5 µm on a paraffin block and mounted into glass slides. These were dried for at least 30 min at 60°C in an oven. The slides were deparaffinized in three changes of xylene (5 min each), 2 changes of 100% EtOH (3 min each), and 2 changes of 95% EtOH (3 min each). Rehydration was performed in dH$_2$O for 1 min. To block endogenous peroxidase activity, 3% hydrogen peroxide was used for 10 min, followed by a rinse in dH2O to remove antigens. For the primary antibody (Anti-Rabbit GFP Abcam ab290, diluted at 1:100 in PBS [pH = 7.1]), 200 mls of the citrate, pH 6.1, antigen-retrieval buffer from Dako (10X concentrate) were used. The buffer was preheated to 80°C in a Black and Decker vegetable steamer for 20 min. The slides were then cooled to room temperature in running tap water (about 15 min). Slides were thoroughly rinsed in water and placed in TBST. After antigen retrieval, 10% normal rabbit serum was applied to the slides and incubated for 60 min at room temperature. Afterwards, they were washed with PBS and the excess was drained. After incubation, Vectastain Elite ABC was used, followed by DAB chromagen (Dako), and incubated for 5 min, followed by washing. All slides were counterstained in hematoxylin for 30 s. Slides were rinsed in tap water until clear and coverslipped.

## Teratoma formation

Chicken and quail iPSC-like cells and control fibroblasts were grown in 6 well plates, detached, and spun down (200g, 5 min). The supernatant was removed, and cells were cleaned and re-spun with PBS (1X, pH: 7.2). The concentration of cells was adjusted to $5 \times 10^6$ cells per ml. 5-week-old male SCID mice (N:NIH-bg-nu-xid; Charles River Laboratories, Raleigh, NC) were used for each experiment. Animals were anesthetized with intraperitoneal injections of ketamine–xylazine (50 and 5 µg/g, respectively) in saline. 100 µl of the cell solution was injected into the mouse testes. Afterwards, the

mice were let to recover from the anesthesia on a heating pad (Kent Scientific). After 5 weeks, the mice were sacrificed, and the testes were removed to assess teratoma formation by histology (H&E, above).

## Chimera formation in chicken embryos

### Cell preparation

GFP labeled iPSC-like cells and fibroblasts were grown, disassociated with Trypsin-EDTA (0.25%) or Pronase (1200 RPM, 8 min), spun down, washed with PBS, spun down again, and re-suspended at 1000 cells per µl, determined by a hemocytometer count.

Injection of cells in embryos: Using a Leica M125 stereo microscope with Xenoworks manipulators (Sutter Instruments, Novato, CA), 1000–5000 cells were introduced into the subgerminal cavity of a 1 day old chicken embryo using a glass micropipette fitted to a microinjector (Sutter Instruments, Novato, CA). Embryos injected with 5000 cells had better incorporation results. Surrogate shells were prepared from organic eggs (*Borwompinyo et al., 2005*) 1 day after being laid. Briefly, the tops of surrogate egg shells were cut off, and eggs were emptied and cleaned with PBS, pH 7.1. Once the experimental (donor) eggs were injected, they were transferred into the surrogate shells. Additional albumin was added if necessary, to completely cover the egg. Finally, the egg was sealed with cling film.

### Incubation

The surrogate egg shells containing injected and control un-injected eggs were placed in an Egg Incubator (P-008Q Bio-type; Showa Furanki Corporation, Japan) set at 39°C. On day 4, embryos were extracted from the surrogate shells and examined under brightfield and fluorescent light. The embryos where then fixed by washing briefly with PBS, and placing the clean embryos in 4% paraformaldehyde buffered with PBS for 3 hr. Finally, embryos were transferred to 70% ethanol and stored at 4°C for immunohistochemistry.

## Chimera formation in fish embryos

Zebrafish were raised as described (*Akimenko et al., 1995*) using standard methods in the Poss Lab zebrafish facility (Duke University). Briefly, GFP labeled control and iPSC-like cells were prepared as described above for chicken cells, and adjusted to a concentration of $10^6$ cells/ml in PBS (pH = 7.1). The cell mixture was placed in a borosilicate glass needle fitted to an Eppendorf CellTram Microinjector. Approximately 100 cells were introduced to blastodisc region of a just fertilized (0 hpf) zebrafish embryo. Embryos were then maintained at 31°C. After 1–3 days, the injected embryos were observed under a fluorescent microscope, to determine GFP labeled cell incorporation. Embryos were then fixed in 4% PFA for 20 min, and placed in 70% ethanol and stored at 4°C for immunohistochemical analysis.

## Acknowledgements

We thank Gustavo Mostoslavsky (Boston University) for providing the STEMCCA cassette vector, Bertrand Pain (Clermont Université, France) for sending us the chicken embryonic stem cells and for very helpful discussions on this study and for maintaining cells, David Burk (The Roslin Institute and Royal School of Veterinary Studies, UK) and Jose Luis Mullor (Hospital Universitario La Fe, Spain) for helping to identify stem cell gene sequences in the zebra finch and zebrafish genomes, respectively, Marguerita Klein (Duke University) for high titer viral generation, Qun Liu for chicken egg injections, Alejandro Aballay (Duke University) for helpful discussions, and Ken Poss and Wen-Yee Choi (Duke University) for assistance with generating the fish chimeras. We also thank Sophie Salama (UC Santa Cruz), David Kohn (University of Michigan), Carol Webb (Oklahoma Medical Research Foundation), Alejandro Aballay, Rebecca Yang (Duke University), and Ken Poss for critical reading of earlier versions of the manuscript. We would also like to thank the University of Puerto Rico Comprehensive Cancer Center (UPRCCC) for the lab space provided to perform some of the experiments.

## Additional information

### Funding

| Funder | Grant reference number | Author |
| --- | --- | --- |
| Howard Hughes Medical Institute | | Erich D Jarvis |
| National Institutes of Health | NIH 5DP1 OD00448-04 | Erich D Jarvis |

| Funder | Grant reference number | Author |
| --- | --- | --- |
| National Institutes of Health | T32NS051156 | Ricardo Antonio Rosselló |
| National Institutes of Health | P30NS061789 | Ute Hochgeschwender |

The funders had no role in study design, data collection and interpretation, or the decision to submit the work for publication.

## Author contributions

RAR, Conception and design, Acquisition of data, Analysis and interpretation of data, Drafting or revising the article, Contributed unpublished essential data or reagents; C-CC, RD, JTH, Acquisition of data, Drafting or revising the article, Contributed unpublished essential data or reagents; UH, Conception and design, Acquisition of data, Analysis and interpretation of data, Drafting or revising the article; EDJ, Conception and design, Analysis and interpretation of data, Drafting or revising the article

# Additional files

## Supplementary files

• Supplementary file 1. (**A**) Controls and experimental groups. (**B**) Conditions used for cell derivation and maintenance. (**C**) Primers used for RT-PCR to amplify and quantify expression of species-specific regions of the genes. (**D**) p-values for the graphs shown in the figures of this study. All comparisons show Tukey's post hoc p-values from an ANOVA test, except for titer (*Figure 2—figure supplement 2*) that shows the overall ANOVA p-value. Bold, significantly different at $p < 0.05$; #, approaches significance. iPSC* = Exogenous expression of factors; iPSCs = Endogenous expression of factors; FB = fibroblast, ESC = Embryonic stem cell, EB = Embryoid body. NA = Not applicable. (**E**) RNA extraction of adult tissue. The contents of the table denote the origin of the adult tissues template used to compare the different levels of gene expression versus embryonic fibroblasts and iPSC-like cells.

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
