## [Decision Letter]

Thank you for choosing to send your work entitled “Mammalian genes induce pluripotent stem cell-like cells in non-mammalian vertebrate and invertebrate species” for consideration at *eLife*. Your article has been evaluated by a Senior editor and 3 reviewers, one of whom is a member of our Board of Reviewing Editors, and one of whom, Shinji Masui, wants to reveal his identity.

The Reviewing editor and the other reviewers discussed their comments before we reached this decision, and the Reviewing editor has assembled the following comments based on the reviewers' reports. The bottom line is that we feel it should be possible to produce a suitably revised manuscript for publication in *eLife*, but we have the following concerns.

1) The failure to produce full iPS-like cells from fly cell lines is potentially interesting but could reflect a technical problem with the cell line used. You should test at least one more *Drosophila* cell source such as BG2 cells (Ui-Tei et al., Apoptosis, 2000) or primary muscle or neuronal cells (Bai et al., Nature Protoc, 2009). In addition, you should test whether changing the reprogramming conditions, for example by using hypoxia or introducing an additional factor such Glis1, allows reprogramming of *Drosophila* cells. If these approaches are not effective, you should discuss the potential reasons in greater depth, such as the ability of *Drosophila* cells to undergo transdetermination (McClure et al., Int J Biochem Cell Biol. 2007).

2) The evidence for pluripotency needs to be strengthened. In Figure 2 the markers you have used, ALP and SSEA1, are markers for pre-iPS cells. It has been reported that only a small proportion of SSEA^+^ cells express the pluripotency markers Oct4 and Nanog (Brambrink et al., Cell Stem Cell 2008). Further concerns are that it is hard to tell from the single clones shown in Figure 2 how efficient the reprogramming is.

A related concern is whether the qPCR data are sufficient to support the claim that the reprogrammed cells are pluripotent, since the levels of induction you observe are modest compared with data from human and mouse cells. Jaenisch and co-workers have shown that in human iPS cells (Soldner et al., Cell 2009), OCT4 and SOX2 are upregulated 1000-fold while NANOG is upregulated 100-fold relative to somatic cells. A 1000-fold increase in Nanog following reprogramming in mouse cells has been observed (Theunissen et al., Current Biol 2011). Your results would be more convincing if you could confirm expression of pluripotency markers by another technique, such as staining with antibodies (if these are effective in the species under investigation) or by Northern blotting.

3) You need to present evidence that the reprogrammed cells continue to express pluripotency makers for a greater number of passages than shown in the current version of the manuscript.

4) In Figure 6 whole mount pictures of control and iPS-GFP should be taken as one picture within a same field. It is otherwise difficult to distinguish background from true expression. One potential reason for the lack of live-born chimeras is that the iPS cells are not fully reprogrammed, as you acknowledge. The injection of GFP-labeled chicken ES cells would be a good control to address this issue.

[Editors’ note: the revised manuscript was re-reviewed, with the following revision requests.]

The reviewers acknowledge that the authors have made substantial revisions. While two of the reviewers are satisfied with the revised paper, one of the reviewers is not. We have read the critical comments carefully and found that most of them are reasonable and they can be summarised as follows:

1) In the qPCR data in Figure 3 (pluripotent markers) of the present manuscript, the authors used fibroblasts (negative cells) as a control, except for the chicken ES cells. In the present manuscript, expression level in fibroblasts was set as 1, although actual expression of Oct4 and Nanog in fibroblasts would be almost zero (undetectable). Comparing to this negative control, the up-regulation of pluripotent markers seen in the presumptive iPSC-like cells of quail, finch, and zebrafish in the present study is quite low. The authors cited the expression data in Takahashi et al. 2006 and [74] Cell, but Takahashi et al. 2006 used the expression of ES cells as controls, also set at 1. Furthermore, the Takahashi and Yamanaka groups additionally characterized the expression of these markers using other methods, western blotting, microarray, DNA methylation, and so on. Thus, these descriptions should be corrected.

2) The expression data on Oct4 and Nanog in their iPSC-like cells except chicken iPSC-like cells are not convincing. The authors should present convincing data on the expression of these pluripotency markers for quail, finch, and zebrafish.

3) While the in vivo data (Figure 6) are some of the most exciting data in this manuscript, the obtained chimeric embryos needed to be seriously judged, although fluorescence of the embryo sometimes causes background. In fact, the immunochemistry slides of Figure 6 need improvement. As they are of too high magnification, we cannot recognize what they are. D and H are almost all positive cells. These data should be improved.

4) The authors could generate teratoma from chicken iPSC-like cells but they did not mention the other iPSC-like lines at all. Their functional data is not strong enough for other iPSC-like lines. It is preferred to present the teratoma formation for non-chicken iPSC-like lines.

---

## [Author Response]

*1) The failure to produce full iPS-like cells from fly cell lines is potentially interesting but could reflect a technical problem with the cell line used. You should test at least one more* Drosophila *cell source such as BG2 cells (Ui-Tei et al., Apoptosis, 2000) or primary muscle or neuronal cells (Bai et al., Nature Protoc, 2009). In addition, you should test whether changing the reprogramming conditions, for example by using hypoxia or introducing an additional factor such Glis1, allows reprogramming of* Drosophila *cells. If these approaches are not effective, you should discuss the potential reasons in greater depth, such as the ability of* Drosophila *cells to undergo transdetermination (McClure et al., Int J Biochem Cell Biol. 2007)*.

We had follow up communications with the reviewers on this request, where we agreed to try generating *Drosophila* iPSC using other cells, but not trying any additional mammalian gene or the putative *Drosophila* homolog (called LMD) to induce iPSC. Following the reviewers’ suggestion, we tried the *Drosophila* Bg2 cell line and we were able to generate iPSC-like proliferating colonies as with the S2 cells. However, the transfection was less efficient and they did not express all the pluripotency markers as found in the induced S2 cells. The results are now included in the paper (Figure 3—figure supplement 3) and discussed in the context of transdetermination, where we propose the S2 cells may already be closer to the stem cell state, making them easier to induce to that state. We thank the reviewers for these suggestions.

*2) The evidence for pluripotency needs to be strengthened. In*
Figure 2
*the markers you have used, ALP and SSEA1, are markers for pre-iPS cells. It has been reported that only a small proportion of SSEA*^*+*^
*cells express the pluripotency markers Oct4 and Nanog (Brambrink et al., Cell Stem Cell 2008). Further concerns are that it is hard to tell from the single clones shown in*
Figure 2
*how efficient the reprogramming is*.

Our understanding of the literature is that ALP and SSEA1 are expressed in both pre and fully developed iPSCs. So, we did not want to conclude, based on the ALP and SEEA1 label alone, that we have pre or fully developed iPSC. This is why this was just one test of many, including assessing other markers, such as Oct4, Nanog, and the endogenous homologs of the stem cell inducing genes, and differentiating the cells in-vivo and in-vitro. It is possible and probable that only a subset of the ALP and SSEA1 labeled cells express Oct4 and Nanog. We have now revised the text and cited Brambrink et al to note that a proportion of induced cells express Oct4 and Nanog, even in mouse cells. We note, however, that our mouse control iPSC had similar ALP and SSEA1 labeling as the other vertebrate species. We also note that even if there is a smaller proportion of iPSC-like cells, like in mice, we are able to generate chimeric avian embryos with them. In terms of single example clones in Figure 2, we had mentioned the efficiency in the submitted paper was 20% of the wells transfected, and that included the mouse cells. With our modified media conditions, we can now obtain efficiencies over 90% of the wells transfected and have now noted this in the revised paper. These findings suggest that the efficiency is independent of mammalian genes inducing iPSC-like state in non-mammalian species, but more dependent on the media conditions used to transform the cells.

*A related concern is whether the qPCR data are sufficient to support the claim that the reprogrammed cells are pluripotent, since the levels of induction you observe are modest compared with data from human and mouse cells. Jaenisch and co-workers have shown that in human iPS cells (Soldner et al., Cell 2009), OCT4 and SOX2 are upregulated 1000-fold while NANOG is upregulated 100-fold relative to somatic cells. A 1000-fold increase in Nanog following reprogramming in mouse cells has been observed (Theunissen et al., Current Biol 2011). Your results would be more convincing if you could confirm expression of pluripotency markers by another technique, such as staining with antibodies (if these are effective in the species under investigation) or by Northern blotting*.

As noted in a separate communication with the reviewers, we believe 10–100-fold changes are not modest but quite large in the context of cellular physiology. The 10–100-fold changes in Oct4, Sox2, and Nanog we find in our control mouse cells and the experimental avian cells are on the same order of magnitude that Yamanaka and many others found in their original studies on mouse cells, and higher by an order of magnitude than they found with their human cells. The two studies quoted by the reviewers that demonstrate a more efficient super-induction method of the stem cell factor genes were published after we had began our study. Therefore, our induction levels are similar to the mouse when using the same method. We have made this clarification in the paper, and discussed the possibility of trying modified methods that have since enhanced induction in mice and human cells, could potentially do so in non-mammalian species. We also note that this level of induction did not prevent the cells from becoming pluripotent in-vivo.

In terms of the method used to test for the presence re-programming gene expression, we did use immunocytochemistry for SSEA1 and RT-PCR for all others. RT-PCR is more sensitive than immunocytochemistry or Northern blots of RNA to quantify gene expression levels. Further, from a technical point of view, demonstrating the expression of re-programming stem cell state genes does not test “pluripotency”. Pluripotency is tested by direct differentiation of the cells into multiple cell types in-vivo and in-vitro. We have now made this distinction clearer in the revised manuscript, so that other readers will not think we are claiming pluripotency from the RT-PCR results. We note that the reviewers did not make a further request for these experiments in our separate note with the editors, which we hope means that the explanations are satisfactory.

*3) You need to present evidence that the reprogrammed cells continue to express pluripotency makers for a greater number of passages than shown in the current version of the manuscript*.

We have now conducted the RT-PCR analyses on the avian cells that have made it past the 5^th^ passage, sampled at the 12^th^ passage, and demonstrate that these cells maintain expression of the endogenous genes several months after being induced and at least 7 passages post-exogenous gene silencing (Figure 3—figure supplement 1).

*4) In*
Figure 6
*whole mount pictures of control and iPS-GFP should be taken as one picture within a same field. It is otherwise difficult to distinguish background from true expression. One potential reason for the lack of live-born chimeras is that the iPS cells are not fully reprogrammed, as you acknowledge. The injection of GFP-labeled chicken ES cells would be a good control to address this issue*.

Taking more pictures of whole mount embryos would require that we repeat those experiments again. We also took the control and iPSC treated animal pictures under identical microscope and camera settings, which were manually set and thus not subject to automatic changes in the exposure by the camera. They were also processed the same. The differences in the GFP fluorescent single was night and day, as seen in the figures. This difference is supported by the striking differences seen in the GFP immuno label images, which can’t be due to background differences. Thus, unless the editors and reviewer feel that repeating this experiment in order to take pictures of the same embryos in the same image is absolutely essential, we would prefer to not have to do so.

The lack of live-born chicken chimeras is because of technical difficulties of keeping embryos alive after surgical manipulation of eggs. We prefer to reserve the publication of live-born chimeras for a separate study on generating transgenic animals with the iPSC-like cells.

In addition, we have since been able to generate in-vivo teratoma formation in mice testes with the chicken iPS-like cells for a longer survival period than the embryos we tested. The generation of these teratomas validates pluripotency in-vivo in the induced chicken cells.

*[Editors’ note: the revised manuscript was re-reviewed, with the following revision requests.*]

*1) In the qPCR data in*
Figure 3
*(pluripotent markers) of the present manuscript, the authors used fibroblasts (negative cells) as a control, except for the chicken ES cells. In the present manuscript, expression level in fibroblasts was set as 1, although actual expression of Oct 4 and Nanog in fibroblasts would be almost zero (undetectable). Comparing to this negative control, the up-regulation of pluripotent markers seen in the presumptive iPSC-like cells of quail, finch, and zebrafish in the present study is quite low. The authors cited the expression data in Takahashi et al. 2006 and*
[74]
*Cell, but Takahashi et al. 2006 used the expression of ES cells as controls, also set at 1. Furthermore, the Takahashi and Yamanaka groups additionally characterized the expression of these markers using other methods, western blotting, microarray, DNA methylation, and so on. Thus, these descriptions should be corrected*.

We thank the reviewers for noting a discrepancy in our response to their previous questions about a comparison of our results with the studies of Takahashi et al. We concur with the reviewers that Takahashi et al used ES cells as their baseline control and we used the starting embryonic fibroblast. We have thus fixed this comparative analysis, and stipulated the similarities and differences between the experiments. Interestingly, we further note that we used mouse and chicken ES cells as controls, and unlike the Takahashi findings we did not find induction of the endogenous genes in the mouse iPS cells above the ES controls, but instead induction to similar levels as the controls, and still well above the starting fibroblast levels. Our comparisons with fibroblasts are, however, consistent with previous work that has compared expression of these genes in somatic cell lines to levels in iPS and ES cells (67). These have shown an induction of 500-1000-fold (for Oct-4, Sox-2, and NANOG), as opposed to our 100–500 fold increase (for Oct-4 and Sox-2) and 10–100 increase in NANOG. We do not know the cause of the differences among studies, but can say that at least our induced non-mammalian cells behaved similar to our induced mammalian, control mammalian, and chicken ES cells. The differences could be technical, in that in the literature different groups using different methods have induced larger or lower levels of gene upregulation (e.g., [67] vs. Theunissen et al., 2011, 1000-fold increase in NANOG).

We also note that our embryonic fibroblasts did not have zero expression of Oct-4 and NANOG, possibly because they are embryonic compared to differentiated fibroblasts, which possibly express these genes at lower levels. Our evidence for above-zero expression is that these two genes were amplified in our PCR reactions after around the 28^th^ cycle.

*2) The expression data on Oct4 and Nanog in their iPSC-like cells except chicken iPSC-like cells are not convincing. The authors should present convincing data on the expression of these pluripotency markers for quail, finch, and zebrafish*.

The expression data on Oct4 and Nanog is 10–50 fold higher than the control fibroblasts in the quail and zebra finch, and this was demonstrated in 5 independent replicates and is highly statistically significant in ANOVA with post hoc test (p<0.001). Although this is not a 100–1000-fold increase, a 10–50 fold increase of gene expression in a cell is still a very large change. In addition, there is no overlap of the expression levels of these two genes with the embryonic fibroblast controls, which we noted above were not zero. In the fish cells the Oct4 expression is also 10-fold above the controls, with a similar significant difference with 5 independent replicates (p<0.001). Nanog, however, we report as not at all upregulated in the fish cells, and we cite papers that show that Nanog is also not necessary or even regulated at high levels in normal fish stem cells. Thus, in the Discussion we suggest that this represents a species difference. Most importantly, these differences between species or lower levels of induced Oct4 and NANOG expression relative to the mice and chicken cells was still sufficient to generate iPS-like quail, finch, and fish cells that can form embryoid bodies. Further, without NANOG upregulation in the fish iPS-like cells, they can still differentiate in fish embryos. And we now show that the quail cells can form teratomas. Thus, we conclude that is it is not necessary for these genes to be induced 100–1000 fold in order for the cells to be pluripotent in-vitro or in-vivo.

*3) While the* in vivo *data (*Figure 6*) are some of the most exciting data in this manuscript, the obtained chimeric embryos needed to be seriously judged, although fluorescence of the embryo sometimes causes background. In fact, the immunochemistry slides of*
Figure 6
*need improvement. As they are of too high magnification, we cannot recognize what they are. D and H are almost all positive cells. These data should be improved*.

Only about 50–60% of the cells for images of panels D and H are labeled for GFP immunocytochemistry. To highlight this difference, we added arrows that show both labeled (white arrows) and unlabeled (black arrows) cells. We also chose different areas with partial and clear staining. This includes whole segments of tissue without GFP label. We also re-took some new pictures to obtain higher resolution images showing the contrast in label.

*4) The authors could generate teratoma from chicken iPSC-like cells but they did not mention the other iPSC-like lines at all. Their functional data is not strong enough for other iPSC-like lines. It is preferred to present the teratoma formation for non-chicken iPSC-like lines*.

In the previous revision, with the iPS-like cells we were able to generate embryoid bodies for all species, in-vivo embryo incorporation for chicken and fish, and a teratoma for chicken, the only species we had tried. We have since tried to generate teratomas with the quail and zebra finch cells, and succeeded with the quail cells. We have added this additional non-chicken functional data in Figure 5 and noted the species difference thus far with zebra finch. We think noting such species differences are important and thus we prefer to keep the finch data in the paper.